

# Remote sensing-aided large-scale rainfall-runoff modelling in the humid tropics

Saúl Arciniega-Esparza[1], Christian Birkel[2,3], Andrés Chavarría-Palma[2], Berit Arheimer[4], José Agustín Breña-Naranjo[5,6]

[1] Hydrogeology Group, Faculty of Engineering, Universidad Nacional Autónoma de México, Mexico City, 04510, Mexico.
[2] Department of Geography and Water and Global Change Observatory, University of Costa Rica, San José, Costa Rica
[3] Northern Rivers Institute, University of Aberdeen, Aberdeen, Scotland.
[4] Swedish Meteorological and Hydrological Institute, Norrköping, Sweden.
[5] Institute of Engineering, Universidad Nacional Autónoma de México, Mexico City, Mexico.
[6] Instituto Mexicano de Tecnología del Agua, Jiutepec, Morelos, Mexico.

*Correspondence to*: Saúl Arciniega-Esparza (sarciniegae@comunidad.unam.mx)

**Abstract.** Streamflow simulation across the tropics is limited by the lack of data to calibrate and validate large-scale
hydrological models. Here, we applied the process-based, conceptual HYPE (Hydrological Predictions for the Environment)
model to quantitively assess Costa Rica's water resources at a national scale. Data scarcity was compensated using adjusted
global topography and remotely-sensed climate products to force, calibrate, and independently evaluate the model. We used a
global temperature product and bias-corrected precipitation from CHIRPS (Climate Hazards Group InfraRed Precipitation
with Station data) as model forcings. Daily streamflow from 13 gauges for the period 1990-2003 and monthly MODIS
(Moderate Resolution Imaging Spectroradiometer) potential evapotranspiration (PET) and actual evapotranspiration (AET)
for the period 2000-2014 were used to calibrate and evaluate the model applying four different model configurations. The
calibration consisted in step-wise parameter constraints preserving the best parameter sets from previous simulations in an
attempt to balance the variable data availability and time periods. The model configurations were independently evaluated
using hydrological signatures such as the baseflow index, runoff coefficient, and aridity index, among others. Results suggested
that a two-step calibration using monthly and daily streamflow was a better option instead of calibrating only with daily
streamflow. Additionally, including PET and AET in the calibration improved the simulated water balance and better matched
hydrological signatures. Thus, the constrained parameter uncertainty increased the confidence in the simulation results. Such
a large-scale hydrological model has the potential to be used operationally across the humid tropics informing decision making
at relatively high spatial and temporal resolution.

## 1 Introduction

Tropical regions differ from temperate regions by larger energy inputs, more intense atmospheric dynamics, higher
precipitation rates, larger streamflow, and sediment yields (Dehaspe et al., 2018; Esquivel-Hernández et al., 2017; Wohl et al.,
2012). Moreover, tropical regions are among the fastest changing environments, with a hydrological cycle pressurized by



population growth (Wohl et al., 2012; Ziegler et al., 2007), land use/cover modifications (Gibbs et al., 2010), and altered
precipitation and runoff patterns (Esquivel-Hernández et al., 2017) due to climate change. Central America, the northern
boundary of the humid tropics, was identified by Giorgi (2006) as the most sensitive tropical region to climate change due to
the location between two major water bodies, the Pacific Ocean and the Caribbean Sea.

Increasing concerns about the effects of human activities and climate change on tropical catchments demand an accurate
quantification of the water balance components in space and time to guarantee the future water resources availability for
ecosystems and socio-economic activities (Esquivel-Hernández et al., 2017; Wohl et al., 2012). Hydrological models have
been widely used to assess the spatio-temporal variability of water resources and to provide insights into potential future
climate and management decisions (Andersson et al., 2015; Xiong and Zeng, 2019).

However, models also implicitly include many uncertainties (Beven, 2012). For example, Birkel et al. (2020) and Dehaspe et
al. (2018) highlighted those hydrological models that are useful to predict streamflow but showed limitations to assess water
partitioning and storage changes required for water management in the humid tropics. Modelling in the tropics is further
hampered by the lack of good quality hydrometric data used to drive models and for calibration (Westerberg and Birkel, 2015;
Westerberg et al., 2014). Moreover, a decrease in hydrological measurements and monitoring networks in many tropical
regions occurred during the last three decades (Wohl et al., 2012), limiting the applicability of hydrological models or reducing
their performance to simulate streamflow in Central America (Westerberg et al., 2014) and South America (Guimberteau et
al., 2012). Model calibration mostly leads to several combinations of parameters with similar streamflow response, i.e.,
equifinality (Beven, 2012; Xiong and Zeng, 2019), and it is therefore desirable to reduce or constrain model parameters
uncertainty. Moreover, some case studies around the world have found that soil model parameters can be relatively insensitive
to streamflow simulations (Massari et al., 2015; Rajib et al., 2018b; Silvestro et al., 2015).

Opportunities exist in form of including additional variables to streamflow for model calibration and validation, providing
more realistic internal hydrological partitioning (Dal Molin et al., 2020; Rakovec et al., 2016; Xiong and Zeng, 2019). The
latter comes at the expense of increased computational cost (Arheimer et al., 2020). For instance, multi-objective calibration
proved useful to reduce parameter uncertainty, but depends on a correct optimization strategy and implementation (Arheimer
et al., 2020; Her and Seong, 2018; Massari et al., 2015; Xiong and Zeng, 2019; Zhang et al., 2018) since the non-linearity
increases the complexity for data assimilation (Massari et al., 2015; Rajib et al., 2018a, 2018b). In addition, hydrological
signatures can improve model realism through the synthesis of many simultaneous catchment processes at different scales
(Arheimer et al., 2020; Sawicz et al., 2011). Despite uncertainties in observed hydrological signatures (Westerberg and
McMillan, 2015) there is potential to identify model weaknesses and to ultimately produce a more well-balanced catchment
representation.

Most hydrological models have been developed since the 1970s to solve different needs at catchment scales (Pechlivanidis
and Arheimer, 2015; Todini, 2007). Nevertheless, water management increasingly requires detailed hydrological information
over larger, aggregated spatial domains instead of a single catchment (Arheimer et al., 2020; Rojas-Serna et al., 2016). Global
hydrological models can serve this purpose but suffer from rather coarse spatial resolution and increased computational cost



(Kumar et al., 2013; Sood and Smakhtin, 2015). Distributed landscape characteristics at large scales such as soil, topography, and land cover can result in complex hydrological models with many calibrated model parameters (Gurtz et al., 1999) and

resulting in greater uncertainty. However, distributed model parameterization based on landscape characteristics also promises the advantage of predicting the hydrological response of ungauged basins (Hrachowitz et al., 2013; Pechlivanidis and Arheimer, 2015). Therefore, the question as to how complex or simple a hydrological model should remain an open science debate considering that simpler models can lead to similar results in comparison with more complex and more highly parameterized models (Archfield et al., 2015; Rojas-Serna et al., 2016).

An alternative to simulate the hydrology at large spatial scales is possible by means of semi-distributed, conceptual hydrological models together with global data of precipitation, evapotranspiration, and soil moisture (Andersson et al., 2015; Brocca et al., 2020). Conceptual models fall in the category between very simple black-box models and physically-based, distributed models maintaining the numbers of parameters more limited with the possibility to still gain insights into the hydrological processes governing a set of neighboring catchments (e.g., Beven (2012) for a model classification). Moreover,

recent hydrological studies implemented data assimilation from remote sensing and global products of soil moisture (Kwon et al., 2020; Massari et al., 2015; Silvestro et al., 2015), snow depth (Infante-Corona et al., 2014), evapotranspiration (Lin et al., 2018; Rajib et al., 2018a, 2018b) and terrestrial water storage (Getirana et al., 2020; Reager et al., 2015) often in combination with conceptual models to reduce or constrain the model parameter uncertainty and to help with model evaluation (e.g., Sheffield et al., 2018). Such an approach needs testing in tropical regions such as Central America, located on the narrow

continental bridge (<40km in places) that connects North with South America. The relatively smaller landmass also results in relatively smaller-sized catchments that quickly convert coarse-scale global products unsuitable for modelling. Additionally, the coarse resolution of climatological model input data is an important source of error, particularly over complex topographical landscapes such as Central America (Maggioni et al., 2016).

Therefore, this paper aims to test the use of the large-scale conceptual but process-based semi-distributed HYPE model

(Lindström et al., 2010) by exploring strategies to improve the regional modelling of tropical data-scarce regions, incorporating different time steps and global gridded products for the complex topographical regions of Costa Rica, in Central America. We, therefore, used the potential and actual evapotranspiration, PET, and AET products, respectively, from MODIS (Moderate Resolution Imaging Spectroradiometer) additionally to streamflow time series to calibrate the model followed by a posteriori independent evaluation of hydrological signatures calculated from these global data sets. The model was calibrated using a

step-wise procedure tracking the most effective strategy to constrain the parameter space and to reduce the model uncertainty. Our specific objectives are the following:

1. Adjust the open-source, conceptual rainfall-runoff model HYPE to simulate Costa Rica's catchment hydrology at the national scale using customized Python-based pre- and postprocessing scripts.
2. Use remotely-sensed global climate data and landscape products to drive and evaluate the model under four different

step-wise calibration strategies.



3. Analyze the effect of remotely-sensed PET and AET data on model calibration and its capability to improve the simulated water balance and matching hydrological signatures.

## 2 Study area and data

The study area corresponds to Costa Rica, located on the Central American Isthmus, between 8 and 11°N latitude and 82 and
86°W longitude. Costa Rica covers ~51,000 km² between the neighboring countries, Nicaragua to the north and Panama to the south. Costa Rica is characterized by an elevation range from 0 to ~3,840 meters above sea level (m.a.s.l.) due to a mountain range of volcanic origin dividing the country from northwest to south-east into the Pacific and Caribbean drainage basins. Notably, the proximity to the two large water bodies (the Pacific Ocean and the Caribbean Sea) differentiates the atmospheric water dynamics resulting in a marked gradient of tropical rainfall patterns east and west of the continental divide (Maldonado
et al., 2013).

Fig. 1.a shows the study area boundaries, the precipitation gauges (blue dots), the monitored catchments (red polygons), and their respective streamflow gauges (black squares), as well as the catchments used within the HYPE model (gray polygons). In situ data consisted of 75 precipitation stations obtained from the National Meteorological Service (IMN in Spanish) containing a minimum length of 10 years of data overlapping the period from 1981 to 2017. This period was selected in order
to compare ground precipitation records with precipitation from global remote sensing products. Moreover, 13 streamflow gauges with daily records from 1990 to 2003 were obtained from the Costa Rican Electricity Institute (ICE). The attributes and climate properties of monitored catchments are shown in Table 1, with catchment areas ranging from 74 to 4,772 km² that cover a total extension of ~10,508 km² (~21 % of Costa Rican territory), with mean catchment elevations ranging from 330 to 2,600 m.a.s.l.
Regarding model simulations, more than 600 nested catchments covering the whole country were delimited using the 30 m Shuttle Radar Topography Mission elevation model (SRTM) (Bamler, 1999) and the terrain analysis toolset from SAGA GIS v.6.4 (Conrad et al., 2015). Several issues were found during the delimitation of catchments on flat terrain, where computed water courses differed from the actual river network. We corrected the computed river network using the vector layers of the main rivers from OpenStreetMaps (OSM), forcing the water courses similar to Monteiro et al. (2018). The final catchments
ranged from 3 to 500 km² with a median value of 65 km² and a main river length from 2.5 to 75 km, and a median value of 15.2 km.

Figure 1.b and Fig. 1.c show the soil type and land use across Costa Rica, respectively. Soil types were derived from SoilGrids (Hengl et al., 2017) (see dataset description in Table 2) and compared to national scale soil maps. Sand content and clay content at 1 m depth were used to classify the soil types from the USDA classification criteria in SAGA GIS tools. Furthermore, in
order to reduce the number of model parameters, only the four most frequent soil types were considered (Fig. 1.b). The predominant soil texture is clay loam covering an extension of ~35,360 km² (69 % of Costa Rica), mainly across low elevation areas. Clay soils cover an extension of ~9,740 km² (~19 % of Costa Rica) and are located mainly along the Pacific basin.





Moreover, in high elevations loamy soils predominate, covering an extension of ~3,800 km$^2$ (7 % of total area). The land use classes were obtained from the Climate Change Initiative Land Cover (CCI LC). Similar to the procedure used for soil types, the land use was reclassified to the four most common categories (Fig. 1.c), where the predominant land uses were tree cover (~65 %) and mosaic cover (~34 %, that includes shrubs, grassland, sparse vegetation, croplands). Urban areas represented less than 0.5 % of Costa Rica.

The climatological space-time series were obtained from remote sensing datasets and global products, described in Table 2. The precipitation grid was obtained from the Climate Hazards Group InfraRed Precipitation with Satellite data (CHIRPS) version 2 (Funk et al., 2015), and the mean daily temperature was obtained from the CPC Global Daily Temperature product provided by the NOAA/OAR/ESRL PSL (https://psl.noaa.gov/). Temperature exhibited low seasonality, with mean values ranging from 27 °C in coastal regions and 20 °C in the central region at around 1000 m.a.s.l. (Esquivel-Hernández et al., 2017). Figure 1.d shows the seasonality of monthly precipitation from CHIRPS using the index proposed by Walsh and Lawler (1981), where lower values (<0.3) correspond to a more uniform monthly precipitation, and higher values (>0.8) indicate that annual precipitation is concentrated over a few months. Such seasonality is widely controlled by air masses that reach Costa Rica at the Caribbean littoral, accumulating more humidity on the Caribbean slope (Sáenz and Durán-Quesada, 2015), shown as dark blue areas in Fig. 1.d. Meanwhile, the humidity along the Pacific basin is highly influenced by the migration of continental air masses of the Intertropical Convergence Zone (ITCZ), which establishes the rainy season in May-June and in September-November (Esquivel-Hernández et al., 2017; Muñoz et al., 2008).

The yearly cycle of wet and dry deviations in the ocean-atmosphere is linked to changes in the sea surface temperature of both the Pacific Ocean and the Caribbean Sea, where the El Niño Southern Oscillation (ENSO) is associated to a decrease of the mean annual precipitation across the Pacific basin, and an increase of precipitation was reported on the Caribbean basin (Muñoz et al., 2008).

Moreover, the cold phase La Niña is the cause of an increase in precipitation in the Pacific basin and a decrease in the Caribbean (Waylen et al., 1996). Overall, the mean annual precipitation averaged for Costa Rica is ~3,000 mm with maxima as high as 9,000 mm, observed in the headwaters of the Reventazón catchment at the northwest of the Talamanca Mountain range and the Caribbean basin (Fig. 1.e). The minimum annual precipitation of 1,200 mm y$^{-1}$ is observed on the northern Pacific basin in the Bebedero and Tempisque catchments.

The rainfall patterns across Costa Rica are reflected in the streamflow responses of catchments on both sides. The daily streamflow tends to be higher in the Caribbean basin (9.2 mm d$^{-1}$ in comparison to 4.2 mm d$^{-1}$ in the Pacific side, computed from observed streamflow records, Table 1), mainly as a consequence of the seasonal climate across the Pacific slope with reduced water availability during the dry season from December to April. Furthermore, the stream length of rivers on the Caribbean slope tends to be longer in comparison with rivers from the Pacific basin (see the river network in Fig. 1.c).

Potential evapotranspiration and actual evapotranspiration were obtained from the Moderate Resolution



**Figure 1**. Study area (a) rainfall stations (blue dots), monitored catchments (red polygons) and sub-basins used in the HYPE model (gray polygons), (b) soil type at 1 m depth from SoilGrids where blue polygons correspond to catchments used for rain correction but not for calibration, (c) major land use categories from CCI Land Cover, (d) precipitation seasonal index





with dark blue colors corresponding to uniform monthly precipitation and yellow colors to more seasonal precipitation regime, (e) mean annual precipitation for the period 1981-2017 from CHIRPS and (f) mean annual actual evapotranspiration from MODIS.

Imaging Spectroradiometer (MODIS) product (Mu et al., 2011) distributed by the Numerical Terradynamic Simulation Group

at the University of Montana, USA, which compared well to few available ground stations in Costa Rica (Esquivel-Hernández et al., 2017). Even though several products of AET and PET are available at higher temporal resolution (such as GLEAM at a daily time step, Miralles et al., 2011), the spatial resolution of these products is at least five times lower than MODIS (~5x5 km, ~25 km$^2$), and since ~70% of our delimited catchments have areas lower than 100 km$^2$, the spatial resolution of the global products plays an important role to capture the spatial variability of the water balance for modelling.

Figure 1.f shows the mean annual AET from MODIS, which spatially ranges from 547 to 1612 mm. The highest AET values were observed at the coast (Caribbean and Pacific). Moreover, the lower AET values overlap with low humidity areas and sparse vegetation (northwestern Costa Rica), as well as higher elevation cloud cover that decreases soil evaporation (Caribbean slope mountain region).

Additionally, soil moisture content at a 10 km spatial resolution retrieved from the Land Parameter Retrieval Model (LPRM)

from the Advanced Microwave Scanning Radiometer 2 (AMSR2) (Teng and Parinussa, 2018) was used for comparison and to explore the scope of incorporating other data sources for calibration (Table 2).

**Table 1.** Physical and climatological properties of the monitored catchments. Streamflow gauges were grouped according to their location on the Caribbean and the Pacific basins.

| Zone | Station | Area [km²] | Elevation [m.a.s.l.] | | | Slope [%] | Prec [mm/year] | EI | AI | Qt mean daily [m³ s⁻¹] |
|---|---|---|---|---|---|---|---|---|---|---|
| | | | Min | Mean | Max | | | | | |
| Caribbean | Cariblanco | 75 | 761.7 | 1850.8 | 2829.5 | 26.5 | 3079 | 0.44 | 0.54 | 9.02 |
| | Oriente | 229 | 586.5 | 1413.7 | 2740.1 | 40.1 | 4202 | 0.32 | 0.42 | 29.90 |
| | Dos Montanas | 660 | 108.5 | 1316.7 | 3191.0 | 37.2 | 3551 | 0.36 | 0.45 | 53.79 |
| | Terron Colorado | 2061 | 18.6 | 734.7 | 2312.5 | 21.3 | 3175 | 0.44 | 0.56 | 149.66 |
| | Guatuso | 242 | 7.8 | 520.6 | 1881.7 | 18.9 | 3968 | 0.35 | 0.45 | 29.38 |
| Pacific | Providencia | 122 | 1365.8 | 2573.0 | 3479.7 | 44.3 | 2750 | 0.50 | 0.68 | 6.68 |
| | Tacares | 201 | 584.3 | 1404.8 | 2723.8 | 17.6 | 2714 | 0.49 | 0.61 | 11.27 |
| | Guapinol | 210 | 178.2 | 1070.4 | 2173.3 | 23.7 | 2879 | 0.49 | 0.66 | 10.13 |
| | Caracucho | 1133 | 60.9 | 1251.0 | 3252.7 | 24.7 | 3091 | 0.43 | 0.56 | 72.92 |
| | El Rey | 656 | 48.6 | 1142.1 | 2501.5 | 36.5 | 2648 | 0.51 | 0.67 | 31.96 |
| | Rancho Rey | 320 | 0.0 | 490.7 | 1902.4 | 15.3 | 2234 | 0.57 | 0.86 | 9.35 |
| | Guardia | 961 | 0.0 | 336.5 | 1898.2 | 11.0 | 1787 | 0.68 | 1.07 | 23.66 |
| | Palmar | 4771 | 0.0 | 1009.2 | 3791.0 | 30.3 | 3176 | 0.41 | 0.55 | 305.45 |




**Table 2**. Remote sensing and global products used in this study.

| Dataset | Variable | Coverage and Resolution | Period | Scale | Data type | Reference |
|---|---|---|---|---|---|---|
| CHIRPSv2.0 | Precipitation (P) | 50°S- 50°N, ~5 km | 1981-present | daily | Merged remote sensing interpolated and calibrated using more than 14,000 rain gauges | Funk et al. (2015) |
| MOD16 | Evapotranspiration (AET) and Potential Evap. (PET) | Global, ~5 km | 2000-2014 | monthly | AET and latent heat flux based on the Penman-Monteith equation incorporated remote sensed MODIS products | Mu et al. (2011) |
| CPC Global Temperature | Temperature (Tmin, Tmax, Tmean) | 89.75S-89.75N, ~50 km | 1979-present | daily | Gridded temperature from 6000 ~ 7000 global stations | https://psl.noaa.gov/ |
| CCI Land Cover | Vegetation cover (Land Use) | Global, 0.3 km | 1993-2015 | annual | Land Cover maps derived from MERIS remote sensing products and classification models | Bontemps et al. (2013) |
| SoilGrids | Silt, sand and clay content | Global, 0.25 km | - | - | Soil properties derived from soil profiles and machine learning | Hengl et al. (2017) |
| SRTM | Land elevation | 30 m | - | - | SAR interferometry | Bamler (1999) |
| LPRM AMSR2 DS D L3 | Surface Soil Moisture (SM) | 10 km | 2012-present | monthly | SM retrieved from the passive microwave remote sensing data from the Advanced Microwave Scanning Radiometer 2 (AMSR2) X-band, using the Land Parameter Retrieval Model (LPRM) | Teng and Parinussa (2018) |

# 3 Materials and Methods

## 3.1 The HYPE model structure and set-up

To simulate the hydrological response of Costa Rican catchments, we used the Hydrological Predictions for the Environment (HYPE) version 5.9, a semi-distributed hydrological model for the assessment of water resources and water quality at small and large scales (Lindström et al., 2010). The HYPE model could be considered as the evolution of the distributed Hydrologiska Byråns Vattenbalansavdelning (HBV) model (Lindström et al., 1997). HYPE was developed by the Swedish Meteorological and Hydrological Institute (SMHI) as the operative model for drought and flood forecasting across Sweden

(Pechlivanidis et al., 2014). Moreover, HYPE was recently applied to other climatic regions (Andersson et al., 2017; Arheimer et al., 2018; Berg et al., 2018; Lindström, 2016; Pugliese et al., 2018; Tanouchi et al., 2019), including a global scale application (Arheimer et al., 2020).



The HYPE model allows simulating the water balance and nutrient fluxes at a daily or sub-daily scale using precipitation and temperature as forcings (SMHI, 2018). The model structure (Fig. 2.a) describes the major water pathways and fluxes, ensuring

mass conservation at the catchment scale, with the model domain divided into sub-catchments. Furthermore, each sub-catchment is divided into the most fundamental spatial soil and land use classes (SLCs) that depend on the classification of soil types, land cover, climate, and elevation, as shown in Fig. 2.b. The SLCs in HYPE provide the capability to predict streamflows in ungauged basins since the parameters that regulate the fluxes and storages are linked to each SLC, with a maximum of three layers of different soil thickness, as shown in (Fig. 2.b). Water bodies such as lakes and rivers may be

considered as a SLC, where lakes can be defined as natural lakes or regulated dams with multiple water outputs. For full details of the HYPE model, see the description by Lindström et al. (2010) and the open-access code references at https://hypeweb.smhi.se/model-water/.

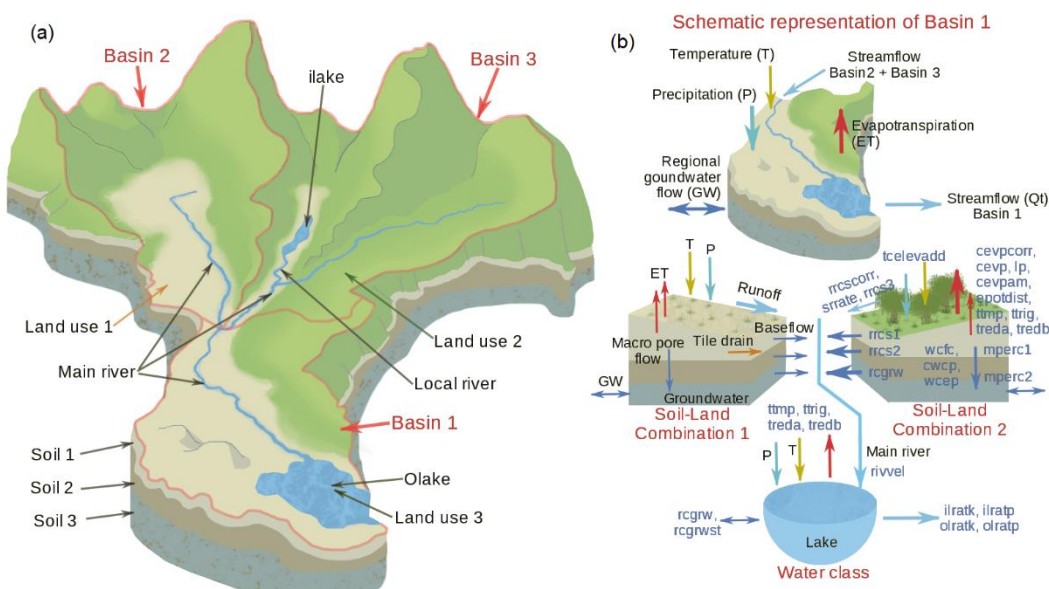

**Figure 2.** Schematic representation of the HYPE model (a) division into sub-basins and local classes according to topography, land use, and soil classes and (b) the model structure of Basin 1 considering two main soil-land combinations and lake properties. In (b), the simulated hydrological processes and variables are shown in black, while parameter names are given in blue. A full description of parameters (in blue) is available in https://hypeweb.smhi.se/model-water/. The *ilake* parameter corresponds to an internal lake, the *olake* parameter to an outlet lake, T is temperature, P is precipitation, ET is

evapotranspiration, Qt is streamflow, GW is groundwater.

For this study, Costa Rica was divided into 605 catchments (Fig. 1.a) with 12 SLCs obtained from the spatial combination of soil types and land cover maps shown in Fig. 1.b and Fig. 1.c, respectively. The outlet lakes (lakes that discharge to downstream





catchments) and internal lakes (lakes that discharge into the main river or tributaries) were set up as different SLCs in order to

consider the water bodies that regulate the streamflow. The largest water body in Costa Rica is the Arenal reservoir, located

in the San Carlos river catchment (Fig. 1.b). The Arenal reservoir is an artificial lake for hydropower purposes with an average

surface area of 87.8 km$^2$ and a depth that ranges from 30 to 60 m. The Arenal reservoir was considered for purposes of this

study as a natural lake since operational rules are confidential.

Soil thickness was considered variable for different SLCs with a maximum depth of 3 m. Furthermore, delimited catchments

were classified according to their elevation and location (Pacific basin and Caribbean basin), applying regional factors to

correct the hydrological behavior of lowland and mountainous catchments with similar SLCs, resulting in six defined regions.

Daily time series of precipitation from CHIRPS and temperature from NOAA for the period 2000-2014 were extracted for

each catchment using GRASS GIS (Neteler et al., 2012), where datasets were resampled to 1 km using the nearest neighbor

criteria and spatially averaged for each catchment. The climatological forcings were resampled due to the small size of some

catchments (area of ~1 km$^2$). Arheimer et al. (2020) recommended the computation of the average of the nearest grids to obtain

the forcings instead of deriving the data from the nearest pixel.

### 3.2 Precipitation correction

Rainfall estimations from satellites are subject to systematic errors that may produce uncertainty in hydrological simulations

(Goshime et al., 2019; Grillakis et al., 2018; Infante-Corona et al., 2014; Wörner et al., 2019). The CHIRPS product already

incorporates a bias correction procedure but uses few and concentrated ground stations in Costa Rica. Therefore, we applied a

linear scaling to further correct the bias found between the product and ground precipitation from 75 available stations across

Costa Rica (Fig. 1.a). The corrected precipitation was estimated as:

$$CHIRPSc(t) = CHIRPS(t) * B,$$ (1)

Where CHIRPSc is the bias-corrected precipitation at time t, CHIRPS is the original precipitation at time t, and BF is the bias

factor. The bias factor was estimated as:

$$BF = \frac{\mu(P)}{\mu(CHIRPS)},$$ (2)

Where μ(P) is the mean of the historical precipitation from ground stations, and μ(CHIRPS) is the mean of the historical

precipitation from CHIRPS. Note that μ(P) and μ(CHIRPS) were computed using the common study period. The simple linear





bias correction was preferred due to the lack of a long common period for all stations to apply more complex methods such as
quantile mapping. Therefore, we used the individual records of more than 60 available stations covering a period from 1980
to 2010 to better capture the complex topography and resulting rainfall patterns.

Some monitored catchments exhibiting higher annual streamflow than annual precipitation could not be corrected due to
groundwater contributions from neighboring catchments (Genereux and Jordan, 2005; Genereux et al., 2002), under-catch at
rainfall gauges (Frumau et al., 2011), and the insufficient number of precipitation stations to correct the CHIRPS database at
a national scale. Nevertheless, errors in climatological data have been found the most common issue for water balance
modelling in Central America (Westerberg et al., 2014; Birkel et al., 2012). In that sense, an additional approach was
implemented to reduce the unrealistic relationship between streamflow and precipitation, which consisted in the creation of
virtual stations at the catchment centroid where the new bias factor was computed as:

$$BF2 = \frac{\mu(Qty) + \mu(AETy)}{\mu(CHIRPSy)}, \tag{3}$$

Where $\mu(Qty)$ is the mean of the annual streamflow for the period 1990-2003, $\mu(AETy)$ is the mean of the annual AET from
MODIS for the period 2001-2014, $\mu(CHIRPSy)$ is the mean of the annual precipitation from CHIRPS for the period 1990-
2014. BF2 adjusts the long-term precipitation volume to ensure that the water balance is preserved, avoiding underestimation
of streamflow and evapotranspiration. Four streamflow gauges in addition to those shown in Table 1 were used to cover more
spatial area in high elevations for the correction of satellite-based precipitation. These streamflow gauges were omitted from
the model calibration-validation procedure due to their shorter records (lower than 7 years). The location of the four streamflow
gauges and their watersheds are shown in Fig. 1.b.

Finally, BF points from precipitation stations and BF2 from virtual points were interpolated using the IDW method with an
exponent value of 2 at the original CHIRPS resolution. The interpolated map of the bias factor was used to spatially correct
the time series of CHIRPS across Costa Rica.

### 3.3 Evapotranspiration and temperature correction

HYPE incorporates four methods for PET estimation (SMHI, 2018). After initial tests, we found that the monthly PET signal
from MODIS in Costa Rica can be reproduced by only using temperature as forcing, where PET is computed as:

$$PET = (cevp * cseason) * (temp - ttmp) * (1 + cevpcorr), \tag{4}$$





Where PET is the daily potential evapotranspiration (in mm), temp is the daily mean air temperature (°C), cevp is an
evapotranspiration parameter that depends on the land use (mm °C-1 d-1), ttmp is a threshold temperature for
evapotranspiration (°C), cevpcorr is a correction factor for evapotranspiration, and cseason is a factor computed as:

$$cseason = 1 + cevpam * sin\left(\frac{2*\pi*(dayno-cevpph)}{365}\right),$$ (5)

Where cevpam is a correction factor, dayno is the day of the year and cevpph is a factor to correct the phase of the sinus
function to correct the potential evapotranspiration (set as zero in this study). To deal with the coarse spatial resolution of the
temperature database (0.5°), a correction factor that depends on catchment elevation was computed (SMHI, 2018):

$$tempc = temp - \frac{tcelevadd*elev}{100},$$ (6)

Where tempc is the corrected air temperature (in °C), temp is the original air temperature (°C), tcelevadd is a calibrated
parameter that corrects temperature (°C 100⁻¹m⁻¹), and elev is the mean catchment elevation (m). Since only few (< 10)
temperature station records were available, a bias correction procedure was not possible, but measured temperature closely
followed the environmental lapse rate (Esquivel-Hernandez et al., 2017).

### 3.4 Model calibration procedure

Figure 3 shows the workflow adopted for model calibration, which involves a qualitative parameter sensitivity analysis to find
the most suitable range of values for the automatic calibration. The initial parameter ranges were obtained from manual
iterations of one parameter at a time to facilitate automatic calibration (Infante-Corona et al., 2014).
We considered four model configurations to analyze the effect of including PET and AET into model calibration:

- Model configuration 1 (M1), calibrated using only daily streamflow (Qt).
- Model configuration 2 (M2), calibrated using first, monthly streamflow and daily streamflow.
- Model configuration 3 (M3), incorporates a calibration using monthly PET and AET, followed by daily streamflow.
- Model configuration 4 (M4), similar to M3 including a calibration using monthly streamflow before the daily streamflow calibration.

The steps are described in Fig. 3. The common period between Qt and PET-AET is relatively short (3 years), resulting in Qt
and PET-AET calibration using different steps. The automatic calibration consisted of a step-wise procedure, where each
model configuration was calibrated for different fluxes (daily Qt, monthly Qt, monthly AET, monthly PET). The parameter





names and initial ranges used for the calibration steps and their configuration are shown in Table 3 and Fig. 2. A final step (not
shown in Fig. 3) consisted in calibrating the curve discharge parameters of the Arenal reservoir using observed water levels.
However, the Arenal infrastructure does not contribute to the downstream basins and has a poor impact on the regional model
calibration. Moreover, as previously stated, the reservoir was simulated as a lake since operational rules are unknown.

The streamflow records were divided into the period from 1991 to 1999 for calibration and from 2000 to 2003 for validation.
The PET and AET calibration period was established from 2002 to 2010 and the validation period from 2011 to 2014. In both
cases, we ran two years prior to calibration for model warm-up. The 13 monitored catchments were used for streamflow
calibration. For PET and AET calibration steps, only the 130 catchments within the 13 monitored catchments were used since
our tests showed that using the 605 catchments did not significantly increase the model performance but increased the
calibration time more than five times. The simulations of the 605 catchments were used to compute the metrics for the
calibration and validations periods.


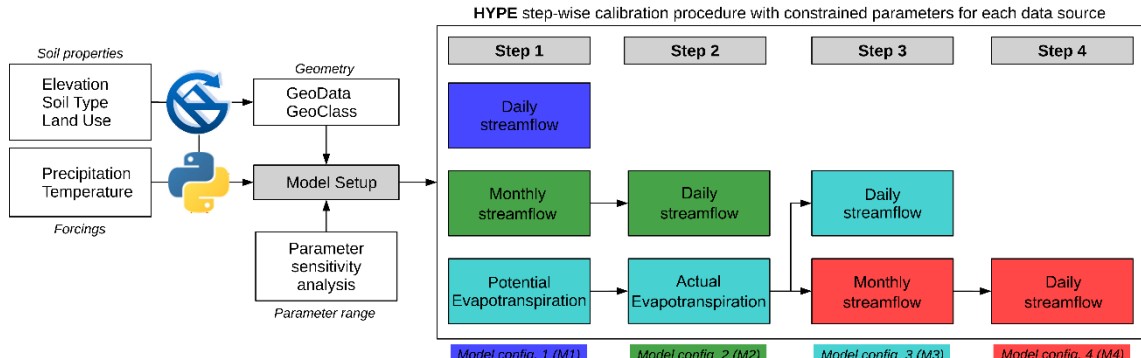

**Figure 3**. Schematic representation of the HYPE model calibration strategy considering a step-wise procedure to constrain
parameters. Four model configurations were established using different data sets or different time scales. From each calibration
step, the 10th and 90th values of the best-fit parameters were used to constrain the parameters of the next step.


A total of 86 parameters were used to build the HYPE model structure consisting in 9 parameters linked to soil types (a total
of 36 parameters considering 4 soil types), 6 to land use (a total of 24 considering the 4 land), 6 for the general structure, 12
for regional correction of PET and T, and 8 for lake discharge. The Monte Carlo (MC) routine for parameter sampling and
sensitivity analysis included in HYPE was used for calibration, and the model configurations were run 10,000 times for each
step, except for M1, which used 20,000 runs to cover more parameter combinations since this configuration only used daily
streamflow. Despite the lower computational efficiency of the MC with respect to other optimization schemes (such as
gradient-based methods), the MC routines are more flexible in accounting for multiple parameters sets in complex models
(Beven, 2006). The 10[th] and 90[th] percentiles of the resulting parameters from the best 100 runs were used to constrain the
parameters for the next calibration step.





**Table 3**. Parameter names and initial ranges of the step-wise parameter estimation for each model configuration. Columns M1, M2, M3, and M4 correspond to parameters optimized during each step of the configuration, and Qt, PET, and AET correspond to the observed time series used to calibrate the model configurations. Qtd means daily streamflow, Qtm monthly streamflow, PETm monthly potential evapotranspiration, and AETm monthly actual evapotranspiration.

| Model configuration | | | M1 | M2 | | M3 and M4 | | M3 | M3 | M4 |
|---|---|---|---|---|---|---|---|---|---|---|
| | | | Step 1 | Step1 | Step2 | Step1 | Step2 | Step3 | Step3 | Step4 |
| Random samples | | | 20000 | 10000 | 10000 | 10000 | 10000 | 10000 | 10000 | 10000 |
| Calibration and validation time step | | | Day | month | day | month | month | day | month | day |
| Variable for objective function | | | Qt | Qt | Qt | PET | AET | Qt | Qt | Qt |
| Calibration period (YY-YY) | | | 91-99 | 91-99 | 91-99 | 02-10 | 02-10 | 91-99 | 91-99 | 91-99 |
| Validation period (YY-YY) | | | 00-03 | 00-03 | 00-03 | 11-14 | 11-14 | 00-03 | 00-03 | 00-03 |
| **Type** | **Process** | **Parameter** | **Initial Rage** | **Step 1** | **Step1** | **Step2** | **Step1** | **Step2** | **Step3** | **Step3** | **Step4** |
| Regional | PET | cevpcorr | -0.35-0 | X | X | | X | | | | |
| | Runoff | rrcscorr | -0.1-0.1 | X | X | | | | X | X | |
| Soil | Runoff | srrate | 0.01-0.3 | X | X | X | | | X | X | X |
| | Runoff | rrcs1 | 0.01-0.3 | X | X | X | | | X | X | X |
| | Baseflow | rrcs2 | 0.001-0.2 | X | X | | | | X | X | |
| | Percolation | mperc1 | 1-100 | X | X | | | X | | | |
| | Percolation | mperc2 | 1-100 | X | X | | | X | | | |
| | Water content | wcfc | 0.2-1.0 | X | X | | | X | | | |
| | Water content | wcwp | 0.01-1.0 | X | X | | | X | | | |
| | Water content | wcep | 0.1-0.8 | X | X | | | X | X | X | |
| Land | PET | ttmp | 1.0-14 | X | X | | X | X | | | |
| | PET | cevp | 0.05-0.4 | X | X | | X | | | | |
| | AET | ttrig | 0.5-1.5 | X | X | | | X | | | |
| | AET | treda | 0.01-1.5 | X | X | | | X | | | |
| | AET | tredb | 0.5-15 | X | X | | | X | | | |
| | Runoff | srrcs | 0.01-0.3 | X | X | X | | | X | X | X |
| General | AET | lp | 0.4-1.4 | X | X | | X | X | | | |
| | PET | cevpam | 0.15-0.4 | X | X | | X | | | | |
| | Temperature | tcelevadd | -0.35 to -0.1 | X | X | | X | | | | |
| | PET | epotdist | 0.5-2.0 | X | X | | X | | | | |
| | Streamflow | rivvel | 0.1-0.5 | X | X | X | | | X | X | X |
| | Runoff | rrcs3 | 0.0001-0.01 | X | X | X | | | X | X | X |



## 3.5 Model calibration and validation using hydrological signatures

CHIRPS product was evaluated with ground records using the False Alarm Rate (FAR, computed with Eq. (7)), Probability of Detection (PD, computed with equation 8), and Threat Score (TS, computed with Eq. (9)):

$$FAR = \frac{falsealarms}{hits+falsealarms},$$ (7)

$$PD = \frac{hits}{hits+misses},$$ (8)

$$TS = \frac{hits}{hits+falsealarms+misses},$$ (9)

Where hits are days with precipitation detected by CHIRPS and ground rain gauges, false alarms are days where precipitation was detected only by CHIRPS, and misses are days where precipitation was detected only by rain gauges.

The model performance was evaluated using the Kling-Gupta Efficiency (KGE) computed as (Kling and Gupta, 2009):

$$KGE = 1 - \sqrt{(r-1)^2 + (\propto -1)^2 + (\beta-1)^2},$$ (10)

$$r = CC = \frac{cov(xo,xs)}{\sigma_o \sigma_s},$$ (11)

$$\propto = \frac{\sigma_s}{\sigma_o},$$ (12)

$$\beta = \frac{\mu_s}{\mu_o},$$ (13)

Where suffixes o and s correspond to observations and simulations, respectively; μ is the mean, x is the time series, σ is the standard deviation, r or CC is the correlation coefficient, α is the agreement between amplitude and β is the bias. KGE was chosen as the objective function for calibration since it equally captures maximum and minimum flows (e.g., Arheimer et al., 2020; Pechlivanidis and Arheimer, 2015; Rajib et al., 2018a; Rakovec et al., 2016; Xiong and Zeng, 2019). Furthermore, other statistical criteria were computed to assess the performance of the model configurations, such as the Pearson correlation coefficient (computed with Eq. (11)), Mean Absolute Error (MAE, computed with Eq. (14)), Nash-Sutcliffe efficiency (NSE, computed with Eq. (15)) and Root Mean Square Logarithmic Error (RMSLE, computed with Eq. (16)):





$$MAE = \frac{1}{n}\sum_{i=1}^{n}|x_s(i) - x_o(i)|,$$ (14)

$$NSE = 1 - \frac{\sum_{i=1}^{n}(x_s(i) - x_o(i))^2}{\sum_{i=1}^{n}(x_o(i) - \mu_o)^2},$$ (15)

$$RMSLE = \sqrt{\frac{\sum_{i=1}^{n}(\log(x_s(i)) - \log(x_o(i)))^2}{n}},$$ (16)

Furthermore, hydrological signatures were calculated to independently assess how well the calibrated model configurations

reproduce different hydrological criteria. Hydrological signatures can be used to increase our understanding of water balance

partitioning, and hydrological similarity across different scales (e.g., Arciniega-Esparza et al., 2016; Beck et al., 2015;

Kirchner, 2009; Troch et al., 2009) and have been applied to improve model evaluation (e.g., Andersson et al., 2015; Arheimer

et al., 2020; Dal Molin et al., 2020; Raphael-Tshimanga and Hughes, 2014; Westerberg et al., 2014). The hydrological

signatures used in this study are shown in Table 4.


**Table 4**. Hydrological signatures used as independent performance evaluation criteria

| Signature | Equation | Description |
|-----------|----------|-------------|
| Mean.Qtd | $\mu = \frac{1}{n}\sum_{i=1}^{n} Qd(i)$ | Mean flow of daily streamflow series |
| Median.Qtd | $m = \frac{1}{2}\left(Qd\left(\frac{n}{2}\right) + Qd\left(\frac{n+1}{2}\right)\right)$ | Median value of daily streamflow series |
| Slope.Qtd | $slope = \frac{Qd_{0.33} - Qd_{0.66}}{0.66 - 0.33}$ | Slope of the flow duration curve |
| CV.Qtd | $CV = \frac{\mu(Qd)}{\sigma(Qd)}$ | Variation coefficient, ratio between mean and standard deviation |
| SC | $SC = \frac{1}{N}\sum_{y=1}^{N}\frac{\sum_{i=1}^{365} Qd(y,i)}{\sum_{i=1}^{365} P(y,i)}$ | Streamflow Coefficient, mean of annual streamflow divided by annual precipitation |
| BFI | $BFI = \frac{1}{N}\sum_{y=1}^{N}\frac{\sum_{i=1}^{365} Qb(y,i)}{\sum_{i=1}^{365} Qd(y,i)}$ | Base Flow Index, mean of annual baseflow divided by annual streamflow |
| AI | $AI = \frac{1}{N}\sum_{y=1}^{N}\frac{\sum_{i=1}^{365} PET(y,i)}{\sum_{i=1}^{365} P(y,i)}$ | Aridity Index, mean of annual potential evapotranspiration divided by annual precipitation |
| EI | $EI = \frac{1}{N}\sum_{y=1}^{N}\frac{\sum_{i=1}^{365} AET(y,i)}{\sum_{i=1}^{365} P(y,i)}$ | Evaporative Index, mean of annual actual evapotranspiration divided by annual precipitation |
| FDC | $sort(Qd)$ | Flow Duration Curve, a plot displaying the statistical distribution of daily streamflow in a decreasing order |



## 4 Results

### 4.1 Remote sensing input data bias correction and evaluation

Comparing precipitation from CHIRPS with annual streamflow and streamflow plus evapotranspiration (assuming long-term balance P-Qt-AET=0) showed underestimated annual precipitation (as shown in Fig. S1.a and Fig. S1.c from the Supplementary Material), leading to unrealistic water balance values. The interpolated bias correction factor (BF, Fig. 4.a) showed overestimated CHIRPS rainfall in blue and underestimations in red. The BF ranged from 0.65 to 1.57 with an average of 1.06±0.14, where the higher disagreements between the ground precipitation and satellite-merged precipitation were

observed along the Pacific basin. The underestimation reached 30 to 35 % in the north of the Gulf of Nicoya and in the southwest of the Providencia catchment. Underestimation of CHIRPS across the Caribbean slope was mainly observed in the Terron Colorado and Cariblanco catchments, with a BF between 1.2 and 1.4. Moreover, the largest overestimation of CHIRPS was observed for the Guanacaste region (BF=0.65-0.8), downstream of the Tacares catchment (BF=~0.8), and to the south-east of Costa Rica (BF=0.8-0.85).

For modelling purposes, we evaluated the temporal synchronicity of rainfall versus streamflow (Fig. 4.b) using cross-correlation between daily streamflow and catchment-scale daily precipitation from CHIRPS, where the x-axis corresponds to the lag time in days. Most of the monitored catchments exhibited the highest correlation within lag time zero, indicating that the hydrological response of catchments tends to occur within the same day. Nevertheless, the Cariblanco and Rancho Rey catchments exhibited a poor correlation (ρ<0.3), which means a lack of synchrony between daily satellite-merged precipitation

and streamflow.

The bias correction improved the annual precipitation where CHIRPSc was consistent with annual streamflow, and the long-term water balance was mostly preserved, as observed in Fig. S1.b and Fig. S1.d. Figure S2 shows the mean absolute error (MAE) for CHIRPS and bias-corrected CHIRPS (CHIRPSc), both with respect to the 75 precipitation stations, where boxplots correspond to the variability of MAE estimated by each point. The average MAE at a daily scale was estimated at 9.4±3.3 mm

for CHIRPS and 9.7±3.7 mm for CHIRPSc, 70.3±29.2 and 67.9±29.1 mm at a monthly scale, and 446.2±308.3 and 331.4±203.3 mm at an annual scale, respectively.

Figure S2.d shows the probability of success and failure of CHIRPSc to detect rainy or dry days with respect to ground stations, where the probability was computed from a single time series merged from the 75 station records. Furthermore, Figure S2.e to Fig. S2.g shows the False Alarm Rate (FAR), Probability of Detection (PD), and Threat Score (TS), respectively. Results

indicated that CHIRPSc detected true rainy and dry days with a similar probability (0.31 to 0.34) compared to the in situ observed rainfall. Whereas the FAR ranged from 0.15 to 0.38 with the larger values (i.e., incorrect detection of dry days as rainy days by CHIRPS) to the southeast, and the PD showed larger values (i.e., better performance of CHIRPS to detect rainy days) on the Pacific basin (median of ~0.69) in comparison with the rain gauges on the Caribbean basin (median PD of ~0.54). The TS showed similar results to PD (Fig. S2.g), with a better performance of CHIRPS on the Pacific basin (median of ~0.53)





than for the Caribbean basin (median of ~0.46). Such low capacity of rainy days detection of CHIRPS on the Caribbean basin

could clearly affect the performance of the hydrological model.

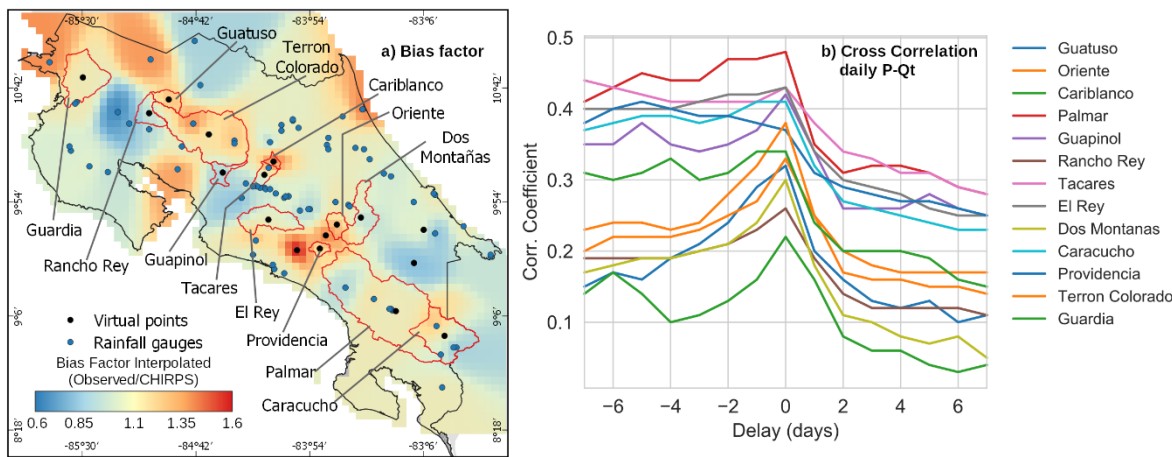

**Figure 4**. Performance of the CHIRPS precipitation product in representing observed rainfall in Costa Rica. (a) interpolated
bias factor where red areas indicate underestimation and dark blue areas overestimation of CHIRPS with respect to ground
stations, and (b) cross correlations between daily precipitation from CHIRPS and observed daily streamflow, where high
correlation values without delay (at zero) indicate that precipitation and high flows tend to occur at the same day. A high
correlation with negative delays indicates that precipitation occurred on average before the streamflow response.

**4.2 Model performance and parameter uncertainty**

Figure 5 shows the comparison of the model configurations' performance for the calibration (dark blue color) and validation
(light blue color) periods. Simulated daily streamflow for the 13 gauged catchments was similar for M1 and M2 during the
calibration period (1991-1999) (Fig. 5.a) with a mean KGE of 0.54 ±0.09 and 0.53±0.08, respectively. Nevertheless, M1
showed a mean NSE of 0.24, twice as high as for M2. Moreover, metrics for M3 and M4 were slightly poorer due to a larger
dispersion across the sample, with a mean KGE of 0.45±0.2 and 0.47±0.17, respectively, and a mean NSE of 0.23±0.2 and
0.21±0.21. For the validation period (2000-2003), the mean KGE decreased by ~0.08, but with similar performance for NSE.
The full statistics are shown in Table S1.

The configuration M2 best reproduced monthly streamflow for the calibration period (Fig. 5.b), with a mean KGE of
0.67±0.11, whereas the configurations M1, M3, and M4 showed a mean KGE of ~0.60, also driven by larger dispersion along
the KGE scale. The four model configurations preserved performance for the validation period, and in some cases, the KGE
even increased, as was the case for the Palmar, Caracucho, and El Rey catchments (not shown). Nevertheless, the Rancho Rey
catchment exhibited poor performance during the validation period (KGE<0) for daily and monthly scales since the four





configurations overestimated streamflow. We present more details for the Rancho Rey that could explain the catchment behavior and its performance in the following sections.

Figure 5.c and Fig. 5.d show the effect of including AET and PET in the calibration steps, and the KGE was computed by aggregating the complete domain (605 nested catchments). The calibration consisted in 130 nested catchments within the monitored catchments. Furthermore, M1 and M2 were plotted for comparison purposes only since these configurations were calibrated only with streamflow. From Fig. 5.c, we observed that simulated monthly AET for the calibration period (2002-2010) improved for M3 and M4 with a mean KGE of ~ 0.49±0.17 with respect to M1 with a mean KGE of 0.29±0.29 and M2

(0.04±0.33). Surprisingly, M1 showed a better performance of simulated monthly PET, with a mean KGE of 0.64±0.09, whereas M3 and M4 showed a mean KGE of ~0.61±0.10, and M2 a mean KGE of 0.43±0.28 (Fig. 5.d). The monthly AET and monthly PET performance were similar for the validation period (2011-2014).

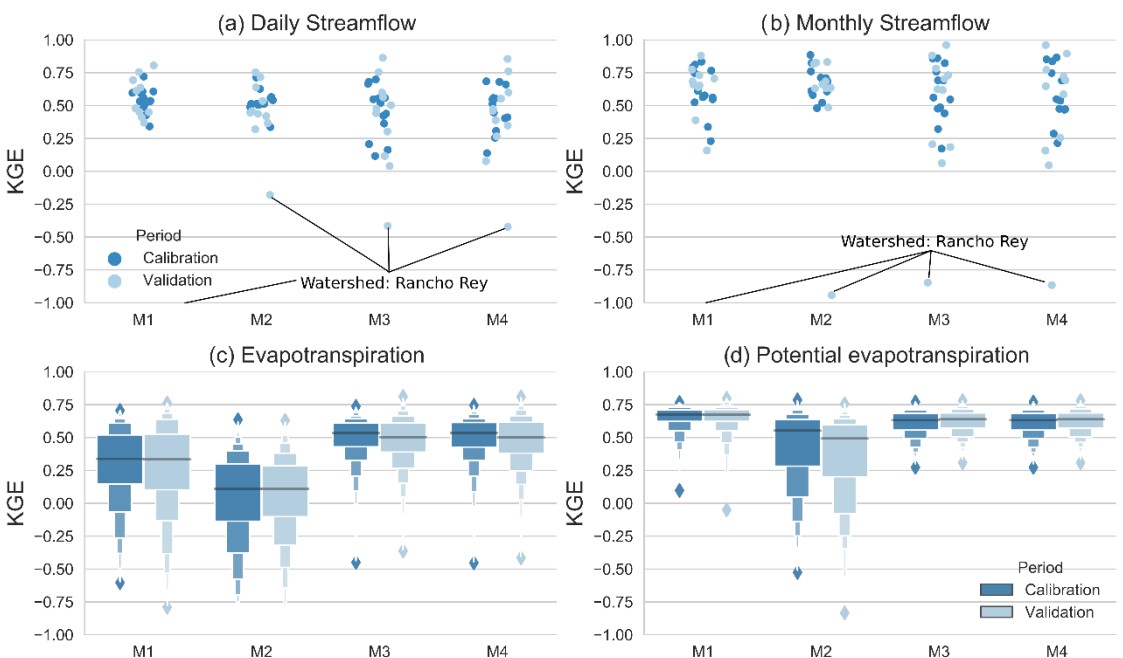

**Figure 5**. The range of KGE values for the calibration (dark blue color) and validation periods (light blue color). (a) the KGE statistical dispersion for daily streamflow, (b) the KGE statistical dispersion for monthly streamflow, (c) boxplots of KGE values for AET, and (d) boxplot of KGE for PET. Streamflow calibration period from 1991 to 1999 and validation period from 2000 to 2003. PET and AET calibration period from 2001 to 2010 and validation period from 2011 to 2014. Since configurations M1 and M2 were calibrated only with streamflow, panels (c) and (d) are for comparison purposes only showing

the effect of including PET and AET in the calibration procedure.



The results from Fig. 6 suggested that the best performances of daily and monthly streamflow for the calibration period (2001-2009) were obtained for catchments in the south Pacific, such as the Palmar, Caracucho, El Rey, and Guapinol catchments

with KGE's higher than 0.55 for daily streamflow and higher than 0.8 for monthly streamflow. Nevertheless, the mid-Pacific basin also resulted in the Tacares and Providencia catchments exhibiting the worst performances for monthly streamflow and the configurations M3 and M4 with KGE<0.3 (Fig. 6.g and Fig. 6.h).

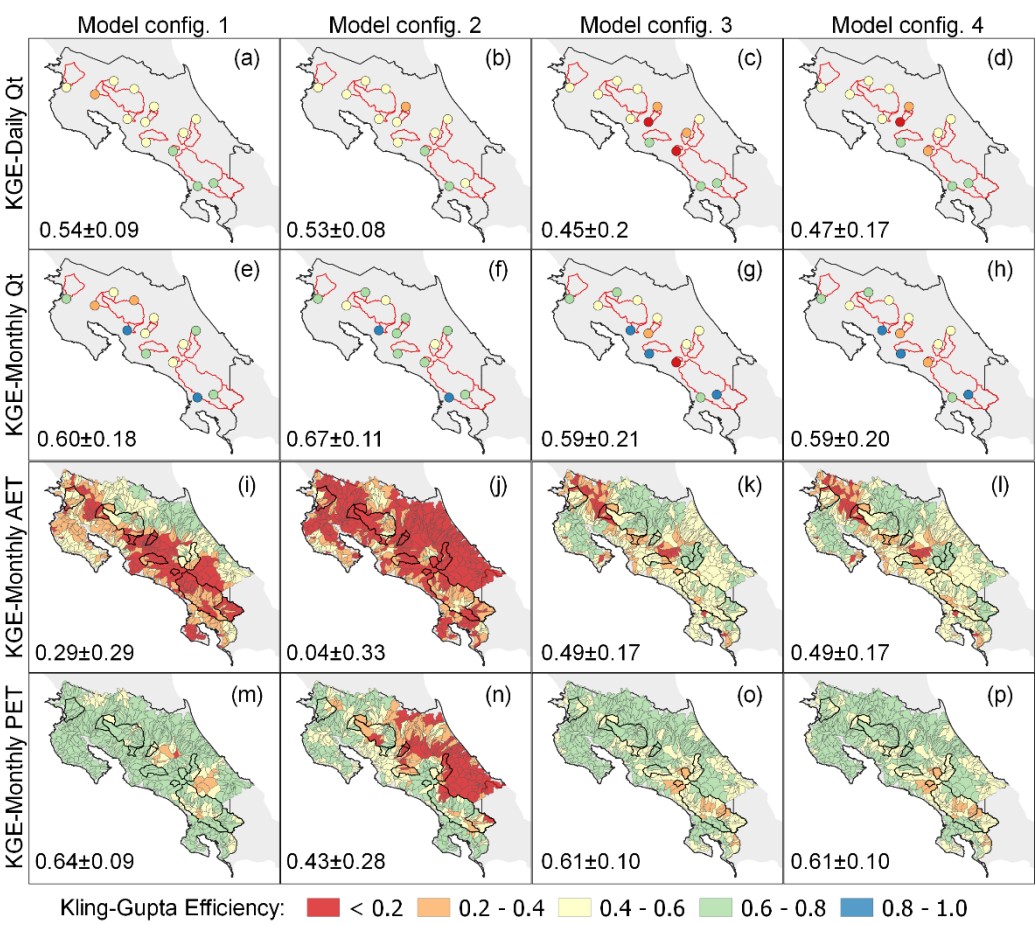

**Figure 6**. Matrix of spatially distributed KGE results for the calibration period (streamflow from 1991 to 1999, PET and AET from 2001 to 2010), where green and blue colors reflect better performance. Configurations M1 and M2 were calibrated only with streamflow. Nevertheless, PET and AET panels are comparative to show the effect of including such variables in the calibration procedure. The mean±std of KGE is shown on the lower left for each panel.






The spatially distributed KGE on the last two panels of Fig. 6 shows the improvement by including AET and PET in the calibration steps (panels (k), (l), (o), (p)), unlike the case of daily and monthly Qt where no significant improvements were observed using the four calibration procedures. The calibrated monthly AET simulated with M1 showed low efficiency (KGE<0.2) for ~182 catchments of the Pacific basin but an acceptable performance (KGE>0.6) for monthly PET. The M2

exhibited poor performance across the simulation domain for AET and low efficiency of PET in the southeast Caribbean. Additionally, M3 and M4 showed similar results with acceptable performance (KGE>0.6) for ~179 catchments, most of them located in the northeast. Surprisingly, the simulated PET with M3 and M4 was similar to PET from M1. The performance of the calibrated water level on the Arenal reservoir was relatively low for all configurations (KGE of ~0.35, not shown), affected by the unknown withdrawals from the reservoir during the driest months (Apr-Jul).

Fig. 7 shows the model configurations parameter uncertainty, where each plot compares the distribution of the parameters from the 100 best-fit simulations resulting from the last calibration step for each configuration. The red dots from Fig. 7 correspond to the optimal parameters used for modelling, where multiple red dots and boxes for each model are shown by soil type and land use.

A large dispersion with a coefficient of variation values (CV=std/mean) larger than 0.35 was observed for runoff response

parameters (srrate, fraction for surface runoff; srrcs, recession coefficient for surface runoff) and baseflow parameters (rrcs1, recession coefficient for uppermost soil layer; rrcs2, recession coefficient for lowest soil layer). The impacts of monthly streamflow on calibration were observed for the general model parameters of rivvel (river velocity) and rrcs3 (deep layer recession coefficient) with constrained posterior parameter distributions for configurations M2 and M4 and larger velocities and baseflow discharge for M2 with respect to M4.

The soil type and land use coverage influence the calibrations' parametrization. M2 and M4 showed constrained distributions of parameters srrate and rrcs1 for clay-loam soil (third class), the most frequent soil type in the monitored catchments (Fig. 1.b). The bottom panel at Fig. 7 shows the spatial distribution of the srrate parameter, with similar values for M2, M3, and M4 and the most frequent soil classes (clay and clay-loam).

The soil parameters that regulate the soil water content (wcwp, wcep) showed similar distributions with the median value of

the fraction of soil water available for evapotranspiration (wcfc). The effective porosity (wcep) was slightly higher for configurations M1 and M2, but the final parameters (red dots) differed between models. Furthermore, for M3 and M4, the parameters lp and cevpam exhibited constrained distributions with a CV of 0.12 and 0.11, respectively. In comparison, M1 and M2 showed CV values of ~0.25 and ~0.28 for lp and cevpam.





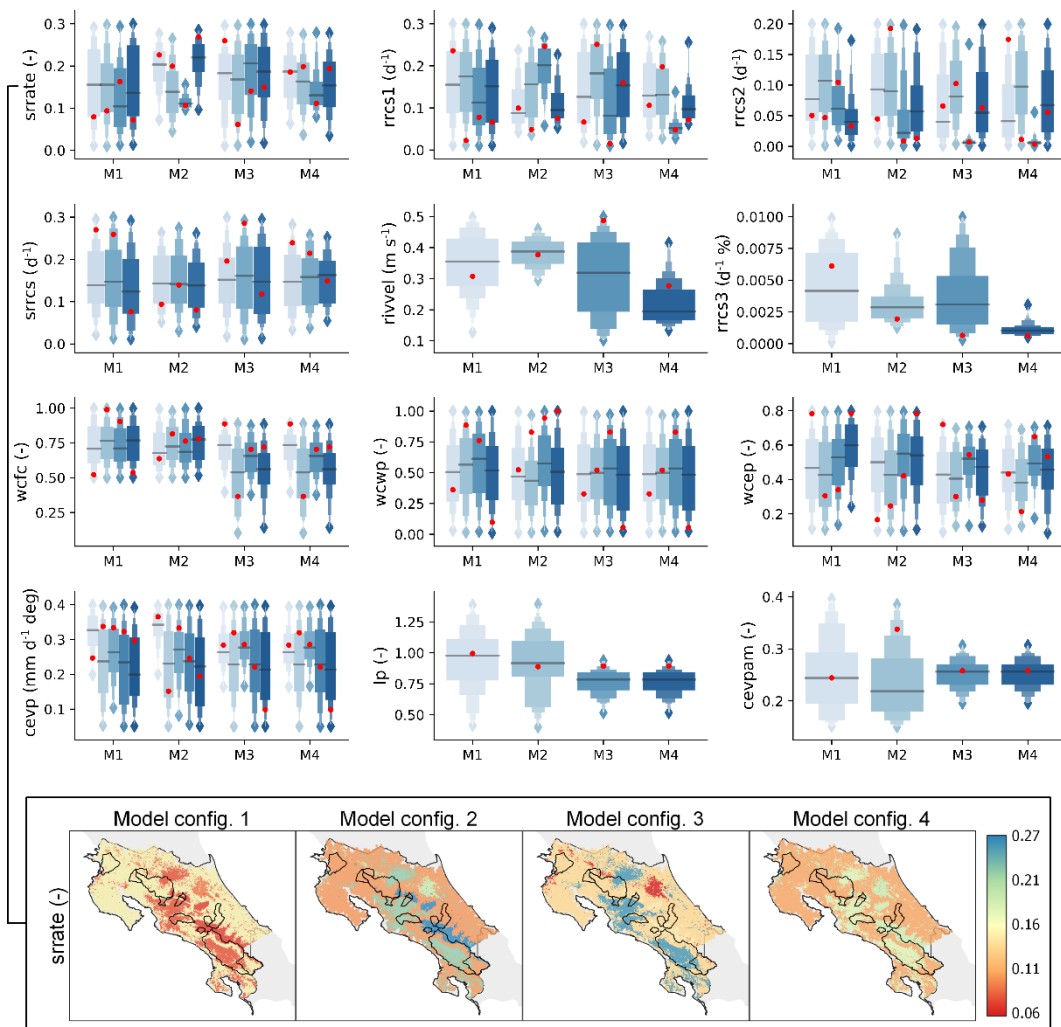

**Figure 7**. A *posteriori* parameter distribution for the 100 best-fit simulations from the last calibration step for each configuration, where red dots correspond to the optimal parameters. Multiple boxes by model configuration correspond to parameters for different soil and land classes. The first two parameter panels correspond to streamflow components, the third panel to water content parameters, and the last panel to PET and AET processes (see Table 3 for reference). For comparison purposes, the bottom panel shows the spatial variability of the best-fit calibrated srrate (-) parameter for each configuration.

## 4.3 Evaluating streamflow simulations and hydrological signatures

The step-wise calibration improved the model performance in different aspects. Fig. 8 shows the comparison of the hydrological simulations for two monitored catchments contrasting the best simulation with the highest KGE performance (Palmar catchment) and the worst simulation with the lowest KGE performance (Rancho Rey catchment).





The Palmar catchment exhibited an acceptable performance (KGE>0.5) for daily streamflow, but all configurations underestimated the highest peak flows during the calibration and validation periods. For the Rancho Rey catchment, the observed highest peak flows were three times larger than simulated peak flows. Underestimation of simulated peak flows was related to the poor capabilities of CHIRPSc to detect heavy storms since observed peak flows were not associated with large precipitation amounts. In Palmar and Rancho Rey catchments, M1 underestimates the low flows by one and two orders of magnitude during the dry season, respectively.

At a monthly scale, streamflow was preserved by the model configurations in several catchments, except for Rancho Rey where simulated streamflow was on average two times larger than the observed streamflow. Such overestimation indicated that the bias factor was insufficient to correct the global precipitation product or large discharge measurement errors. Furthermore, all configurations reproduced the seasonality of AET and PET from MODIS, but M3 and M4 underestimated the AET and PET in Palmar with a good performance for AET in Rancho Rey. Moreover, simulated monthly soil moisture (SM) content was independently compared with the catchment average soil moisture content from the LPRM product for the period 2012 to 2016. The simulated SM for M1 followed the seasonal behavior of the LPRM product in the Palmar catchment, matching the absolute LPRM % SM content. The LPRM product uses SM from the upper 5 cm against the 50 cm of the upper layer defined for all the model configurations. However, all model configurations have shown a high correlation (>0.7) in both catchments (Palmar and Rancho Rey), matching the seasonality.

The observed and simulated flow duration curves for all monitored catchments are shown for the period 1991-2003 in Fig. 9. The M1 (blue line) underestimated the median and low flows in several catchments (Guardia, Rancho Rey, Guatuso, Terron Colorado, Caracucho, El Rey, Guapinol) in comparison with other configurations. The M2 (green line) exhibited the best performance for median and low flows, whereas M3 (cyan line) and M4 (red line) showed similar results to M2. Higher efficiencies for median and low flows were obtained for catchments that exhibited higher cross-correlation with precipitation (Fig. 4.b), as was the case for Palmar, Caracucho, and El Rey, among others.

The simulated and observed hydrological signatures are shown in Fig. 10, where simulations covered the period 1991-2014, and observations covered different periods depending on available records. The simulated long-term mean annual water balance (Prec-Qt-AET) was mostly closed in all catchments (~0 mm), with average values of -1.80 mm (M1), -0.62 mm (M2), -0.79 mm (M3), and -1.17 mm (M4), hence the water balance was significantly improved when compared to the observed water balance. Indeed, observed (Prec-Qt-AET) yielded values going from -800 to 600 mm. Such variability in observed water balances may be related to the short common period of data but also due to the discrepancies between data sources (in situ, interpolated and merged, remotely sensed), as can be observed in Fig. S3 from the Supplementary Material. Figure S3 shows how the long-term water balance using the observed data differs from the Budyko curve in all monitored catchments (Westerberg et al., 2014), while the simulations fitted the theoretical curve.

The spatial distribution of baseflow indices (BFI) derived from M2, M3 and M4 exhibited similarities with respect to the observations. Simulated BFI showed an overwhelming groundwater contribution to streamflow with relatively similar average values of 0.70, 0.69, 0.68, and 0.74 for the model configurations M1, M2, M3, and M4, respectively.

**Figure 8**. Simulated versus observed time series for catchments with the best streamflow KGE performance (left column, Palmar watershed) and the worst streamflow simulation (right column, Rancho Rey watershed). Black lines correspond to observations where streamflow is shown from 1990 to 2003, MODIS AET and PET from 2000 to 2014, and the LPRM soil water content (SM) from 2012 to 2016.



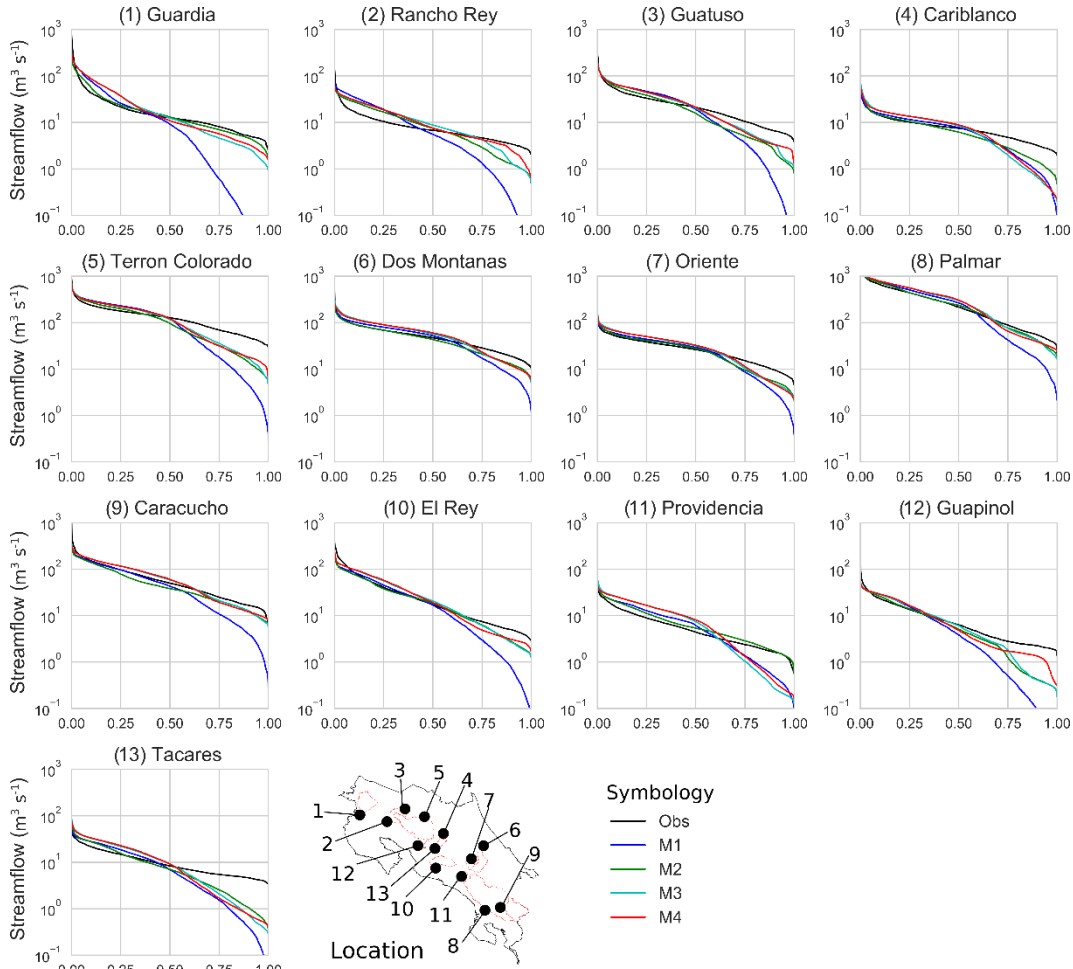

**Figure 9**. A comparison of observed flow duration curves (FDC) as a hydrological signature and the matching simulated FDC for each model configuration. The simulated period was from 1991 to 2003.


Larger differences were observed in the northwest and southwest when comparing the BFI of M1 with respect to other configurations, whereas M4 resulted in larger contributions of baseflow to streamflow in coastal areas of the Caribbean. Similar spatial patterns were obtained for the streamflow coefficient (SC=Qt/Prec), with low values in the drier northwest and higher values for catchments that receive more rainfall (Fig. 1.e). In contrast, the M2 indicated that a lower amount of precipitation became streamflow, with an average value of 0.49, in comparison with M1, M3, and M4 that showed medians of 0.52, 0.57, and 0.57, respectively. Moreover, M1 and M2 followed the spatial patterns of observed SC due to their higher streamflow performance.





The Evaporative index (EI) and Aridity index (AI) were similar for M3 and M4 due to similar model parameters. The spatial distribution of observed EI from MODIS was reproduced by M3 and M4, whereas AI spatial patterns were preserved by M1.

Besides, M1 and M3-M4 showed similar spatial patterns for EI and AI across the north, but differences were observed in the south, where M1 indicated lower water availability attributed to higher evaporative ratios (higher EI). M2 simulated the driest catchments with an average value of EI and AI of 0.50 and 0.63, respectively, whereas M1 showed median values of 0.47 and 0.59, and M3-M4 values of 0.42 and 0.51.

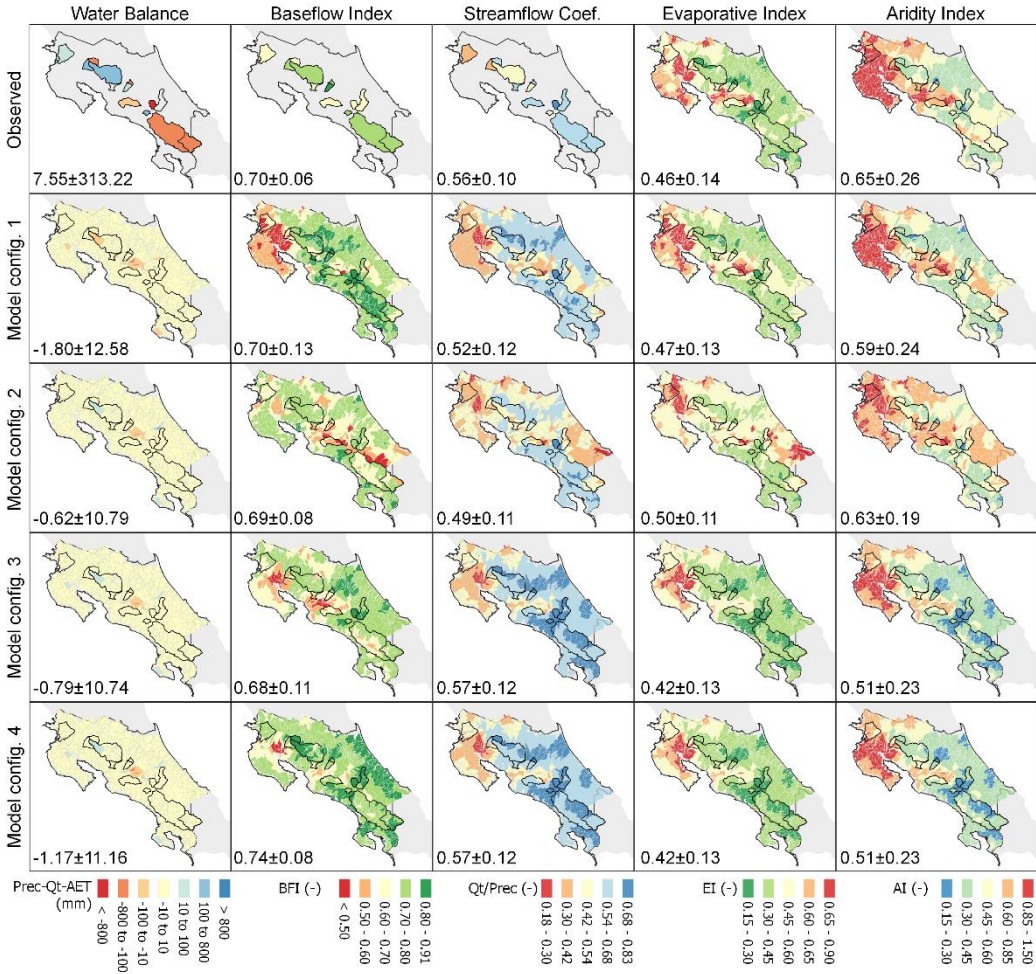


**Figure 10**. Observed and simulated spatial distribution of hydrological signatures. BFI: Baseflow Index (Baseflow / Streamflow), EI: Evaporative Index (AET / P), AI: Aridity Index (PET / P). Signatures from observations were obtained for the periods 2000-2003 (Balance), 1991-2003 (BFI, SC), and 2002-2014 (EI, AI). Signatures from simulations were obtained for the period 1991-2014, and the mean±std are shown on the lower left for each panel.




## 4 Discussion

### 4.1 Remote sensing and global products as model forcings and calibration series

Daily precipitation from CHIRPS was preferred over other global precipitation products because of a relatively higher spatial resolution and good performance across different climates and biomes (e.g., Bayissa et al., 2017; Ullah et al., 2019; Zambrano-Bigiarini et al., 2017). Nevertheless, CHIRPS showed a large bias and more rainy days with respect to ground precipitation across Costa Rica (Fig. 4.a). The results suggested that at large scales, the precipitation bias was compensated since the mean bias factor (BF) was ~1, but underestimation of precipitation was observed in mountainous regions as wells as large overestimations in the drier northwest (Fig. 4). Similarly, results from Chile by Zambrano-Bigiarini et al. (2017) indicated that an evaluation and even correction of global products is necessary and may be related to the poor density of ground gauges used to generate CHIRPS in those regions, especially across regions with steep topography (Funk et al., 2015). The latter authors also reported an underestimation during extreme rainfall events and an overestimation during rainy days.

Our simple, linear bias correction of CHIRPS showed better performance at monthly and annual scales and solved the water balance inconsistencies of most catchments (Fig. 10). However, the cross-correlation between daily precipitation and streamflow remains unchanged by the bias correction. Not surprisingly, our results showed that catchments with highly correlated streamflow and daily precipitation exhibited better performance than catchments with low correlations. Several studies highlighted those meteorological forcings are the largest source of uncertainty in hydrological modelling (e.g., Arheimer et al., 2020; Dal Molin et al., 2020; Lin et al., 2018; Wörner et al., 2019), whereas more complex bias correction techniques (e.g., quantile mapping) may improve the results (e.g., Goshime et al., 2019; Wörner et al., 2019), but, the lack of matching daily streamflows with precipitation inputs and severe rain events might persist. Nonetheless, Infante-Corona et al. (2014) suggested that global products can achieve better streamflow simulation results than sparse ground precipitation data, whereas Westerberg and Birkel (2015) found that in situ precipitation in Costa Rica may require corrections to achieve better model results.

The global CPC temperature dataset used was not bias-corrected due to the lack of sufficient in situ measurements. Temperature can introduce large errors in hydrological simulations if used for the estimation of potential evapotranspiration and actual evapotranspiration (Andersson et al., 2015). We corrected the temperature data set using elevation and a lapse rate parameter (Eq. (6)) during the calibration steps. The corrected temperature closely followed the environmental lapse rate of 6ºC temperature decrease per 1000 m elevation gain and significantly improved the model configurations performance. Global evapotranspiration estimates showed differences compared to ground estimations in different regions due to the influence of, e.g., irrigation, vegetation dynamics, and uncertainty in climatological forcings (Pan et al., 2019; Velpuri et al., 2013). We used the PET and AET from MODIS16 for calibration and evaluation due to reported good performance for different applications around the world (e.g., Lin et al., 2018; Mu et al., 2013; Pan et al., 2019; Rajib et al., 2018b; Tang et al., 2011; Velpuri et al., 2013). Nevertheless, MODIS AET has shown poor performance at point-scale in different regions (e.g., Liu et al., 2015; Weerasinghe et al., 2019), but better performance when aggregated at the catchment scale (Velpuri et al., 2013).





Additionally, Wohl et al. (2012) recognized that dense vegetation and frequent cloudiness in the humid tropics are challenges for satellite monitoring of AET. Unfortunately, the low density of eddy covariance towers and lack of comparative studies for tropical climates are limiting factors to validate MODIS16 in Central America. Nonetheless, among the few existing studies available, Esquivel-Hernandez et al. (2017) compared the MODIS16 PET product against 10 Priestley-Taylor station data derived PET estimates in Costa Rica and found a deviation of less than 20%.

## 4.2 HYPE performance in data-scarce tropical catchments

Simulated daily streamflow showed reasonable performance using the four model configurations, where M2 (calibrated Qt monthly + Qt daily) improved low flows simulations in comparison with M1 (Fig. 9). Moreover, our configuration using a step-wise calibration for Costa Rica resulted in improved streamflow performance compared to the global model by Arheimer et al. (2020). The main shortcomings of the four configurations were underestimated peak flows by two or three times (Fig.
8), but such errors were associated with the precipitation product rather than the model capabilities. The spatial comparison of streamflow simulations indicated that catchments in the southwest performed best (KGE>0.5, Fig. 6) compared to other areas. The southwestern Pacific is characterized by a moderate precipitation seasonality (Fig. 1.d) with a low bias of the precipitation product (Fig. 4.a) compared with the tropical climate gradient of the dry to humid tropics in Costa Rica. Furthermore, we found overestimated monthly streamflows in the drier northwestern region of Costa Rica (Rancho Ray and Guardia catchments,
Table S3). Previous studies have noted that HYPE overestimates streamflow in dry environments (Arheimer et al., 2020) so this could support such findings. Finally, the streamflow overestimation could also be related to the precipitation bias of CHIRPSc and possibly to the nonuniform spatial distribution of our streamflow observational sample, with more "wetter" catchments used for calibration.

Testing of PET and AET in calibrations (M3 and M4) resulted in significant improvements of model realism (internal water
partitioning, also indicated by more sensitive parameters lp from Fig. 7), but a slight decrease in streamflow performance (Fig. 5 and Fig. 6). Such multi-objective calibration trade-offs were previously observed by, e.g., Zhang et al. (2018). Larger improvements were obtained for AET simulation of M3 and M4, whereas M1 (calibration only daily Qt) showed similar performance of PET in both periods (calibration and validation). The worst PET and AET simulations were observed for M2 since the monthly aggregation ignores an accurate representation of spatial water partitioning to match the monthly hydrograph
(Rajib et al., 2018b). Our results also suggested that low flows were improved using PET and AET for calibration (Figure 9), where FDC exhibited an average RMSLE (Eq. (16)) value of ~0.5±0.22 compared to 1.1±0.53 from M1, constraining vertical fluxes and regulating discharge from soil layers (Massari et al., 2015; Rakovec et al., 2016). The constrained a posteriori parameter distributions related to evapotranspiration processes (cevp, lp, cevpam) indicated increased parameter sensitivity by MODIS PET and AET calibration. The soil water content parameters (wcfc, wcwp, wcep), however, only slightly improved
from PET and AET calibration (Fig. 7). Additionally, including monthly Qt into the calibration routine also constrained parameters related to soil layer discharge (srrate, rrcs1, rrcs3) for the most frequent soil types on the monitored catchments.





Such results highlight that remotely sensed PET and AET are useful to constrain some parameters and that the combination of data sources representing different modeled hydrological processes helps constrain model uncertainties, particularly for large-scale domains (Rajib et al., 2018b). Despite the partly different calibration periods used due to limited data availability, similar

record lengths (8 years calibration and 3 years in validation) resulted in consistent results from M1 to M4. Finally, the LPRM soil moisture product captured the soil moisture seasonality and showed promising capabilities for regional model calibration and correction in Costa Rica.

Despite the generally reasonable performance of our model configurations, we found some issues when comparing our results with previous efforts, such as, Birkel et al. (2012), who modeled the streamflow in the Sarapiqui River basin (data not used in

this study for calibration) with the HBVlight model (Seibert, 2005) for the period from 1983 to 1991 and obtained a NSE of 0.74 after a rainfall correction of the underestimated observed precipitation. Our configuration M4 resulted in a NSE of -5.6 due to rainfall underestimation (Frumau et al., 2011). In contrast, our model configuration M4 reflected improvements with respect to the global product of Arheimer et al. (2020). For example, we obtained a KGE of 0.73 for the streamflow simulation of the Palmar catchment, in comparison to the global product showing a KGE lower than 0.3. Such results reflect that more

data is required to improve the spatial variability of the streamflow response at the local scale.

### 4.3 Independent model evaluation using hydrological signatures

A hydrological model useful for water management should be able to mimic streamflow's intra-annual variability and to realistically represent the large-scale physical processes of the water partitioned by vegetation interception and the soil matrix

into evapotranspiration and discharge (Arheimer et al., 2020; Kwon et al., 2020; Pechlivanidis and Arheimer, 2015; Rajib et al., 2018b; Rakovec et al., 2016; Xiong and Zeng, 2019). We, therefore, independently evaluated the four configurations using a range of hydrological signatures (Table 4) similar to Westerberg and McMillan (2015) in an attempt to single out the sought-after well-balanced model for use in decision making.

Significant spatial variations in hydrological signatures were observed between M1-M2 and M3-M4, but implementing a

spatial calibration of AET improved the representativeness of the more complex large-scale climate gradient (Lin et al., 2018; Rajib et al., 2018a). The model configurations M3 and M4 better reproduced the spatial variability between the Pacific and Caribbean basins and the north-south gradient of the AI and EI, similar to Esquivel-Hernandez et al. (2017). Furthermore, the resulting hydrological signatures of M3 and M4 were consistent with previous small catchment scale studies that showed that runoff coefficients tend to be larger than the evaporative index (Dehaspe et al., 2018; Gómez-Delgado et al., 2011). Results

also suggested that the event streamflow response is dominated by quick near-surface soil water discharge (Dehaspe et al., 2018), with streamflow being fed by groundwater during dry periods resulting in BFI values exceeding 0.7 (Birkel et al., 2012). In contrast to Westerberg et al. (2014), who calibrated Central American catchments using FDC information, we used the observed FDCs as an independent hydrological signature (Fig. 9). The configurations M2 to M4 outperformed M1 supporting the notion that only streamflow used for calibration is not enough to produce a well-balanced model.






**5 Conclusions**

This study is the first attempt to apply the process-based, conceptual rainfall-runoff HYPE model at the national scale of Costa Rica (~600 simulated catchments). Due to the lack of detailed observational data available in Costa Rica, as in most parts of the world's tropics, we used different global topography, soil, land use products, daily streamflow from 13 gauges, the bias-

corrected global precipitation product CHIRPS (with 75 ground stations) and remotely sensed MODIS16 PET and AET products to improve the performance of HYPE in a step-wise calibration procedure towards a well-balanced model useful for water resources management. The calibrated model configurations were independently evaluated using a suite of hydrological signatures. We summarize our main findings here:

- Bias was observed in precipitation from CHIRPS, with underestimation in mountainous regions and overestimation
in the driest region with around 1,000 mm of annual rainfall in Costa Rica.

- CHIRPS showed ~10% more days with rainfall in comparison with ground precipitation but could not capture extreme rainfall events, which ultimately impacts streamflow simulation.

- Our bias correction procedure using the linear-scaling technique reduced the annual water balance inconsistencies (Prec<Q+ET) by 25%. Still, a more complex methodology is required to improve the daily precipitation depth and
timing.

- The temperature could efficiently be used with an elevation correction; nevertheless, a higher resolution temperature product or downscaling approach would improve the many micro-climates across the complex topography in Costa Rica.

- HYPE successfully reproduced major processes (evapotranspiration, runoff, baseflow discharge) of tropical
catchments in Costa Rica, where we obtained acceptable performance for daily streamflow (KGE from 0.4 to 0.6) and good performance for monthly streamflow (KGE from 0.6 to 0.9) with best-fit results for PET and AET of KGE=0.6 and KGE=0.5, respectively.

- Model calibration using monthly and daily streamflow (M2) improved the performance of the low flows in comparison to only daily streamflow (M1) calibration, where the average RMSLE of FDC was computed as 0.42±0.22
for M2 compared to 1.14±0.53 from M1.

- Remotely-sensed PET and AET constrained the soil type and land cover parameters associated with the evapotranspiration process.

- Including PET and AET in calibration (M3 and M4) slightly decreased the overall streamflow performance (average KGE of 0.47±0.17 compared to 0.54±0.09 from M1) at the expense of an improved and more well-balanced median
and low flow (average RMSLE for FDC of ~0.5±0.22 compared to 1.1±0.53 from M1) simulation and evapotranspiration water partitioning.



- Simulated hydrological signatures (aridity index, evaporative index, baseflow index, streamflow coefficient, flow duration curve) differed for each calibrated model, but configurations M3 and M4 more realistically mimicked the spatial distribution of all tested hydrological signatures.


We conclude that M3 and M4 were promising model configurations to quantitively assess water resources in Costa Rica. Improvements to these models could be achieved by incorporating more independent data into the calibration process, such as soil moisture and groundwater level and storage data. However, all global products crucially depend on an evaluation and even correction, which needs observational in situ data. Nonetheless, we hope to have provided a way forward towards a large-sale

operational hydrological model for the humid tropics of Costa Rica and potentially in other humid regions of the world.

**Data availability**

The hydrological simulations of model configuration 4 (M4) represent the HYPE for Costa Rica version 1.0 dataset (HYPE CR 1.0) freely available online (https://doi.org/10.5281/zenodo.4029572) and through an interactive web app visualization

tool (https://zaul-ae.gitbook.io/oacg-hidrologia/).

**Author contribution**

All the authors contributed to the study's conception and design. The data collection was made by Saul Arciniega-Esparza and Christian Birkel. The data processing and the model construction were made by Saul and Andrés Chavarría-Palma. Results

were analyzed by Saul and Christian. Berit Arheimer and Agustín Breña-Naranjo contributed to the methodology review and discussion of the model results. Christian and Agustín critically revised the work, and all the authors approved the final manuscript.

**Competing interests**

The authors declare that they have no conflict of interest.

**Acknowledgments**

Saúl Arciniega-Esparza was supported by the CONACYT graduate scholarship program and the *Programa de Maestría y Doctorado en Ingeniería* at UNAM. Christian Birkel would like to acknowledge funding from the Observatorio del Agua y

Cambio Global (OACG) under UCR grant ED-3319. The authors also thank the Center for Geophysical Research (CIGEFI)





and Dr. Ana Maria Duran at UCR for sharing meteorological station data. We thank Alejandra González Hernández for her support in editing and translating the manuscript.

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
