# Peer review of "Remote sensing-aided rainfall-runoff modelling in the tropics of Costa Rica"

_Hydrology and Earth System Sciences, 2021_

## Author Response (AR1)

Mexico City, Dec 15, 2021

Yi He
Editor
Hydrology and Earth System Sciences

Dear Editor,

We enclosed our revised manuscript entitled "Remote sensing-aided rainfall-runoff modelling in the tropics of Costa Rica" along with our point-by-point responses to the helpful comments provided by yourself and two anonymous Reviewers.

We have fully addressed the review comments in the attached response and have revised the manuscript
accordingly. We made a number of changes to the manuscript (marked in blue) in response to the comments and are confident that these changes now better explain how this analysis is insightful to the HESS readership and wider hydrological community.

In the following, we explain how (i.e. our reply starting the line with "*R:*") and where (i.e. line numbers) each point of your comments was incorporated in the revised manuscript. We hope that this
new version proves to be of interest to you and the reviewers and that it is worth being considered for publication in Hydrology and Earth System Sciences.

Best regards,

Saul Arciniega-Esparza (on behalf of all co-authors).

**Response to Editor**

E1: Both reviewers commented on the title. You have proposed to revise to "Remote sensing-aided large-scale rainfall-runoff modelling in the tropics of Costa Rica". I feel the revised title can be misleading given the rainfall-runoff modelling was not carried out on large-scale catchments. The catchments are of small to medium sizes. I think what you meant by "large-scale" refers to the remote sensing data sets except the CPC T dataset which is station based. I suggest you revise your title to avoid mis-understanding. My suggestion would be "Rainfall-runoff modelling in the tropics of Costa

Rica by using global datasets".
*R: Following your recommendations, we changed the title to "Remote sensing-aided rainfall-runoff modelling in the tropics of Costa Rica".*

E2: L120 - were delimited, L122 - delimitation of catchments. I think you meant "were delineated" and "delineation of catchments". Please consider revising your terminology.
*R: Thanks for your comment. This point was corrected in Line 126.*

E3: Some or all of the global datasets require you to acknowledge the use of their data. please consider adding to your Acknowledgments.
*R: Thanks for your suggestion. This point was included in the revised version.*

After reading the title, I expected to be presented with a modelling study covering larger areas of the humid tropics. I was thus surprised to find that the manuscript only discusses the case study of Costa Rica when reading the abstract. Thus, I suggest to change the title and exchange "humid tropics" with "Costa Rica".

*R: Following your recommendations, we changed the title to "Remote sensing-aided rainfall-runoff*
*modelling in the tropics of Costa Rica".*

Line 121 states that delineation of the catchments was performed using "the terrain analysis toolset from SAGA GIS". Were the standard settings used?
*R: Thank you for this comment. The algorithms and parameters used in SAGA GIS are described in*
*Lines 128-131 of the revised manuscript.*

The description of the 4 calibration strategies and the associated schematic in Figure 3 left me somewhat confused. Looking at the figure, I assumed that M2 was a stepwise calibration in which a first iteration calibrated against monthly streamflow, followed by a second calibration against daily
streamflow. I thus wonder what the "first streamflow" in line 307 refers to. Furthermore, the color coding in Figure 3 left me wondering how M2 and M4 differ from each other and why M4 was similar to M3. The schematic would be clearer if a 4th row could be added, so that each row represents one calibration scheme.
*R: Thanks for this comment. Figure 4 was modified accordingly to your comments.*

Both NSE and KGE values are presented for comparing the performance of the 4 calibration strategies with each other. In line 437 a values of KGE < 0 are deemed to be poor and in lines 474 and 476, values of KGE > 0.6 are said to be acceptable. How is the choice of these ranges justified? As Knoben et al.
(2019) show, even negative KGE values could present an improvement over using the mean flow as a predictor. At the same time, there is no guarantee that KGE > 0.6 is linked to an improvement over a specific benchmark. While the given values clearly show which of the methods provides an improvement over the other, it remains unclear how good the performance actually is. This is particularly relevant in lines 516-521 where an acceptable performance of KGE > 0.5 is linked to both
underestimated high and low flows. I would thus like to see a prupose-based KGE benchmark specified against which the results can be compared.
*R: Thank you for raising this important point. In this version, we evaluated the performance of the different HYPE model setups using multiple performance metrics (KGE, Pearson Correlation Coefficient, MAE, NSE) to improve the calibration description, as is shown in Lines 445, 453, 456, 473,*
*478, 489, 491. Additionally, we implemented a clearer comparison in Figure S3 from the supplementary material.*

**Technical corrections**

Line 274: The abbreviation IDW needs to be defined.
*R: The abbreviation IDW has been defined in Lines 284-285 in the revised version.*

Figure 5: Please extend the y-axis so that the values for Rancho Ray M1 become visible as well.
*R: Thanks. We modified Figure 5 following your suggestions.*

All figures: Unfortunately, the colour scheme used is often not colour-blind friendly. Particularly the lines in Figures 8 and 9 are barely distinguishable. Also, the colour gradient green-yellow-red (e.g. in Figure 1f) or the multicolour gradient (e.g. Figures 4a, 6) generate maps which are very hard to read. I thus suggest switching to a different colour scheme and to use different line shapes (dotted, dashed) to further improve the readability.
*R: Thank you for this suggestion. We modified the figures to improve the visualization using different color schemes more friendly for color-blind readers.*

1. There is a lack of connection between the supplementary section and the main text. For instance, when the authors introduced the CHIRPS product (~L240), they could link it to Fig. S1 to have a clear picture of the improvement. Another example are tables S1 and S2, which are not mentioned anywhere
in the text but would be a useful reference in the discussion section where the authors discuss these hydrological signatures for all models.

*R: Thank you for pointing out that. In this version, we referred to the additional supplementary materials to improve the description of the results and the discussion, as is shown in Lines 251, 452, 473, 571, 652, 696-698.*

2. The abstract needs to be improved by including some of the nice statistics and results from the paper that quantify the improvements. Around L23, the authors talk about the hydrological signatures and that using both daily and monthly streamflow is better than just using the daily flows. However, it is not clear by how much.
*R: Thanks for your suggestion. In this version, the abstract included more details about the model configurations and comparisons of the metrics computed.*

3. The authors do not specify which model configuration is the baseline (which I assume is M1). Furthermore, while they present performance statistics, it is unclear if these differences are statistically
significant to merit the additional data. Moreover, when they discuss the time-series analysis and the differences between the models, they do so in a descriptive manner to quantify it better. For instance, using a distance metric to evaluate series similarity to the observed data. See DOI: 10.1016/j.rse.2011.06.020 for a summary of some useful metrics. My suggestion would be to make plots of the Mahalanobis distance rather than presenting the original time series (or in addition to Fig.
8).

*R: Thank you for raising this important point. We clarified the statistical improvement in the model performance with respect to the baseline (M1) in Lines 535-540, 553, 660-665. Moreover, we modified Figure 8 to better explain the differences between model configurations.*

4. I believe that the first objective should be merged into the other objectives. Running the model (independently of the computer language used) is a trivial objective as it is met from the start of the project.

*R: Thank you for this suggestion. In this version, we merged the first two objectives into a single objective (see Lines 104-106).*

5. The authors need to explain how they did the catchment extraction in GRASS by providing additional detail into the used parameters. Also, they need to explain the IDW method in the methods section, define the acronym, and add a reference.

*R: In Lines 128-131 we explained the parameters used for catchment extraction using SAGA GIS, and*
*in Line 285 we included the corresponding information for IDW method.*

6. The authors need to improve Fig. 3; interpreting it is confusing. Perhaps it would be best to have it with 4 rows rather than arrows, even if there is a degree of repetition.

*R: Thanks. We modified Figure 3 following your comments.*

7. Can the authors modify the presentation of the 86 parameters in L331? It is hard to understand; I would suggest presenting the numbers in parenthesis as the main parameter numbers and then elaborating on how many were linked to soil types, land
cover, etc.

*R: Thanks for your suggestion. Parameters were described in a more precise way in Lines 345-347.*

8. Can the authors add box plots of the other statistics as supplementary? It is hard to visualize them as isolated numbers. Again, can the authors perform tests of significance on the statistics to determine a significant difference between them?

*R: Thanks for your suggestion. We converted Table S1 as Figure S3 to show boxplot in the supplementary materials section.*

9. Can the authors mention what the criteria for defining a KGE of 0.5 as acceptable were?

*R: Thank you comment. We implemented a clearer description and discussion on this issue based on the*
*other performance metrics (KGE, Pearson Correlation Coefficient, MAE, NSE) used for comparison purposes in the supplementary material, as is shown in Lines 445, 453, 456, 473, 478, 489, 491. Additionally, we implemented a clearer comparison in Figure S3 from the supplementary material.*

10. Around L605, the authors mention that the corrected temperature improved model performance. The
authors need to quantify this performance increase.

*R: Thank you for this suggestion. Line 629 shows the general improvement after including temperature correction in the model setups' simulation.*

11. The authors mention that the streamflow overestimation can be related to a precipitation bias in
CHIRPSc. However, from Fig. S1, this does not seem to be the case.

*R: Thank you for raising this important point. We found that precipitation overestimation persisted in drier environments despite the bias correction. The overestimate was associated with the lack of ground precipitation records to correct the CHIRPS product in headwater catchments such as Rancho Rey. Our Fig. S1 in the supplementary section shows that, in many cases, differences between the water balance*
*fluxes (P, ET, Q) were reduced. We clarified this point in Lines 666-659.*

12. When discussing model improvement, please quantify it. The authors mention in L635 that M3 and M4 showed better and more realistic results but failed to quantify the improvement. Moreover, from Fig. 10, it seems that even though KGE was higher for M3 and M4, M1 was able to reproduce the actual
spatial distributions of PET and AET better, overlapping more with the observed ranges.

*R: Thank you for the comment. Only to clarify, M1 showed high performance for PET but a lower performance for ET in comparison with M3 and M4 (shown in Figure 5). In Lines 472, 488-489, 660-665, we extended the discussion about PET and AET comparisons.*

13. In the discussion section, the authors mention that adding PET and AET to the calibration improved model representativeness and link earlier studies. The authors need to also link this assertion to their study, which is one of their objectives.
*R: Thank you for this suggestion. This is included in Line 699.*

14. Due to missing tests, I do not see how the authors can conclude that M3 and M4 are better configurations since the statistical significance of the differences has not been evaluated. And in fact, for a lot of the variables, it seemed that M1 performed adequately well compared to M3 and M4. The authors can further support the increased accuracy of M3 and M4 by their link to the FDC information.
*R: Thank you for your suggestion. In this version, we computed two statistical tests described in Lines*
*394-395. We describe the results obtained with these tests in Lines 660-665, 739-740. Moreover, Figure 8 was modified to clarify the improvements in streamflow simulation.*

       15. Finally, I suggest adding ": A case study in Costa Rica." to the title since it was the only region analyzed in the manuscript.
*R: Following your recommendations, we changed the title to "Remote sensing-aided rainfall-runoff modelling in the tropics of Costa Rica".*

       Around L54, the authors mention the opportunities from including additional variables. Please, specify which variables or give a few examples.
*R: We include some examples of additional variables to improve hydrological partitioning in Lines 58-59.*

       **Technical corrections:**

       Around L56, the authors mention that more realistic hydrological partitioning comes at the expense of increased computational cost. Can the authors quantify the time penalties involved?
       *R: In Lines 60-61 we corrected this statement.*

Around L77, do the authors mean simple bucket models? Any model can be a black-box model.
       *R: Thanks for your comment. Line 83 included this correction.*

       Around L87, the authors mention that the coarse spatial resolution of the climatological data is an important source of error. Can the authors mention the related uncertainty in the data? (i.e., how much
of the model error is associated with the coarse spatial resolution).
       *R: Thanks for your comment. We added some corrections around this statement in Lines 92-95 to explain the errors associated with remote-sensing precipitation.*

L135, the authors mentioned that they merged land covers. Can the authors include how much each
merged class contributed to the overall classification?
*R: This description was added in Lines 145-149 of the revised manuscript.*

L159, please remind the reader what both sides are.
*R: Line 179 includes this correction.*

L175, please quantify the statement; how well did MODIS compare with the ground data? State r2 or
another statistic.
*R: Thanks for your comment. Lines 190 and 640 report metrics computed by previous authors.*

L209, can the authors mention how they chose the soil layer thickness?
*R: Thanks for your comment. Due to the lack of accurate soil thickness maps in Costa Rica, we
considered a maximum soil thickness of 3 m in forest land cover and a minimum of 2 m in bare land
cover, following the recommendation in the model configuration shown by Arheimer et al. (2020). This
point is clarified in Lines 237-238.*

*Arheimer, B., Pimentel, R., Isberg, K., Crochemore, L., Andersson, J. C. M., Hasan, A., and Pineda, L.:
Global catchment*
*modelling using World-Wide HYPE (WWH), open data and stepwise parameter estimation. Hydrology
and Earth System Sciences Discussions, 1–34. https://doi.org/10.5194/hess-24-535-2020, 2020.*

L245, the correction factor appears as B in equation one; it appears as BF and BF2 in equations 2 and 3.
*R: Thanks for your detailed review. We modified this error in equation 1.*

L266, from the text, it is somewhat ambiguous if y refers to each year or the whole period.
*R: Thanks for your comment. We clarified the explanation of equation 3 in Lines 276-277.*

L288, the authors should mention that the parameters for correction are part of a monte Carlo simulation
and are set to the ranges in Table 3.
*R: Thanks for your suggestion. We included this point in Lines 309-310.*

L290, sine function.
*R: Thank you. We corrected this error in Line 300.*

L320, can the authors justify why only two years were used as a warm-up?
*R: Thanks for your suggestion. The warm-up period was established as two years before the calibration
period in order to avoid issues with initial conditions of water content in soil layers, rivers, and
reservoirs. We tested from 1 to 3 years, and results did not change from two to three years. We included
this description in Lines 332-333.*

L361, please remind the reader which time series.
      *R: Thanks. In the revised paper, we will include an explanation of which series are compared.*

      L375-379, this information should appear in the introduction.
      *R: Thanks for your suggestion. In the revised paper, we will move this sentence to the introduction*
*section.*

      L405, can the authors normalize the MAE by the mean precipitation? Doing so would help the reader to understand the relative magnitude of the MAE.
      *R: Thanks for this great suggestion. Figure S2 from supplementary was modified to show the*
*normalized MAE, and Lines 419-423 now describe such results.*

      L415, please, specify how they affect the performance.
      *R: Thanks for your suggestion. This description was included in Lines 656-659, where we explain the implications of unsynchronized peak flows due to precipitation errors.*
      L532, from Fig. 9, it seems that all the models underestimated the real flows to some extent. Is this due to CHIRPSc?
      *R: Thanks for your comment. As you mention, precipitation from CHIRPS is an important factor of error in our results. In Lines 656-659 of the discussion section, we explained how catchments from the*
*Pacific slope showed higher performance in comparison with catchments from the Caribbean slope, related to the performance of CHIRPS to detect rainy and dry years on both slopes.*

      L575, can the authors increase the border thickness of the catchments of Fig. 10? It isn't easy to see them.
*R: Thanks for your suggestion. Figure 10 was modified following your comments.*

      L619, is the deviation a positive or negative bias?
      *R: Thanks for your suggestion. This point was clarified in Line 640.*

L644, what do the authors mean by increased parameter sensitivity?
      *R: Thanks for your comment. We changed this sentence to clarify the implications of MODIS PET and ET on model parameters (see Lines 672-675).*

      L650, Can the authors comment why none of the models at the best performing catchment could
reproduce the decrease in water content between 2014-2015?
      *R: Thanks for your comment. We assume that do you refer to the water content of Rancho Ray catchment in Figure 8. In this case, Rancho Ray showed the poorest performance of all catchments evaluated, which we will better highlight. The lower capacity to reproduce soil water content by most model configurations is related to the precipitation overestimation that stores water in the soil buckets.*
*We improved Figure 8 to show the performance of streamflow simulations, and we extended the description of such performance issues in Lines 656-665.*